# LEARNING COUNTERFACTUALLY INVARIANT PREDICTORS

## ABSTRACT

We propose a method to learn predictors that are invariant under counterfactual changes of certain covariates. This method is useful when the prediction target is causally influenced by covariates that should not affect the predictor output. For instance, this could prevent an object recognition model from being influenced by position, orientation, or scale of the object itself. We propose a model-agnostic regularization term based on conditional kernel mean embeddings to enforce *counterfactual invariance* during training. We prove the soundness of our method, which can handle mixed categorical and continuous multivariate attributes. Empirical results on synthetic and real-world data demonstrate the efficacy of our method in a variety of settings.

## 1 INTRODUCTION AND RELATED WORK

Invariance, or equivariance to certain transformations of data, has proven essential in numerous applications of machine learning (ML), since it can lead to better generalization capabilities Arjovsky et al. (2019); Chen et al. (2020); Bloem-Reddy & Teh (2020). For instance, in image recognition, predictions ought to remain unchanged under scaling, translation, or rotation of the input image. Data augmentation is one of the earliest heuristics developed to promote this kind of invariance, that has become indispensable for training successful models like deep neural networks (DNNs) Shorten & Khoshgoftaar (2019); Xie et al. (2020). Well-known examples of certain types of "invariance by design" include convolutional neural networks (CNNs) for translation invariance Krizhevsky et al. (2012), group equivariant CNNs for other group transformations Cohen & Welling (2016), recurrent neural networks (RNNs) and transformers for sequential data Vaswani et al. (2017), DeepSet Zaheer et al. (2017) for sets, and graph neural networks (GNNs) for different types of geometric structures Battaglia et al. (2018).

Many real-world applications in modern ML, however, call for an arguably stronger notion of invariance based on causality, called *counterfactual invariance*. This case has been made for image classification, algorithmic fairness Hardt et al. (2016); Mitchell et al. (2021), robustness Bühlmann (2020), and out-of-distribution generalization Lu et al. (2021). These applications require predictors to exhibit invariance with respect to hypothetical manipulations of the data generating process (DGP) Peters et al. (2016); Heinze-Deml et al. (2018); Rojas-Carulla et al. (2018); Arjovsky et al. (2019); Bühlmann (2020). In image classification, for instance, we want a model that "would have made the same prediction, if the object position had been different with everything else being equal". Similarly, in algorithmic fairness Kilbertus et al. (2017); Kusner et al. (2017) introduce notions of interventional and counterfactual fairness, based on certain invariances in the DGP of the causal relationships between observed variables.

Counterfactual invariance has the significant advantage that it incorporates structural knowledge of the DGP. However, enforcing this notion in practice is very challenging, since it is untestable in real-world observational settings, unless strong prior knowledge of the DGP is available. Inspired by problems in natural language processing (NLP), Veitch et al. (2021) provide a method to achieve counterfactual invariance based on distribution matching via the maximum mean discrepancy (MMD). This method enforces a *necessary, but not sufficient* condition of counterfactual invariance during training. Consequently, it is unclear whether this method achieves actual invariance in practice, or just an arguably weaker proxy. Furthermore, the work by Veitch et al. (2021) only considers discrete random variables when enforcing counterfactual invariance, and it only applies to

specific, selected causal graphs. To overcome the aforementioned problems, we propose a general definition of counterfactual invariance and a novel method to enforce it. Our main contributions can be summarized as follows:

- Based on a structural causal model (SCM), we provide a new definition of counterfactual invariance (cf. Definition 2.2) that is more general than that of Veitch et al. (2021).

- We establish a connection between counterfactual invariance and conditional independence that is provably *sufficient* for counterfactual invariance (cf. Theorem 3.2).

- We propose a new objective function that is composed of the loss function and on the flexible Hilbert-Schmidt Conditional Independence Criterion (HSCIC) Park & Muandet (2020), to enforce counterfactual invariance in practice. Our method works well for both categorical and continuous covariates and outcomes, as well as in multivariate settings.

## 2   PRELIMINARIES AND BACKGROUND

**Counterfactual invariance.**   We introduce structural causal models as in Pearl (2000).

**Definition 2.1** (Structural causal model (SCM)). *A structural causal model is a tuple $(\mathbf{U}, \mathbf{V}, F, \mathbb{P}_{\mathbf{U}})$ such that $\mathbf{U}$ is a set of background variables that are exogenous to the model; $\mathbf{V}$ is a set of observable (endogenous) variables; $F = \{f_V\}_{V \in \mathbf{V}}$ is a set of functions from (the domains of) $\mathsf{pa}(V) \cup U_V$ to (the domain of) $V$, where $U_V \subset \mathbf{U}$ and $\mathsf{pa}(V) \subseteq \mathbf{V} \setminus \{V\}$ such that $V = f_V(\mathsf{pa}(V), U_V)$; (iv) $\mathbb{P}_{\mathbf{U}}$ is a probability distribution over the domain of $\mathbf{U}$. Further, the subsets $\mathsf{pa}(V) \subseteq \mathbf{V} \setminus \{V\}$ are chosen such that the graph $\mathcal{G}$ over $\mathbf{V}$ where the edge $V' \to V$ is in $\mathcal{G}$ if and only if $V' \in \mathsf{pa}(V)$ is a directed acyclic graph (DAG).*

We always denote with $\mathbf{Y} \subset \mathbf{V}$ the outcome (or prediction target), and with $\hat{\mathbf{Y}}$ a predictor for that target. The predictor $\hat{\mathbf{Y}}$ is not strictly part of the SCM, because we get to tune $f_{\hat{\mathbf{Y}}}$. Since it takes inputs from $\mathbf{V}$, we often treat it as an observed variable in the SCM. As such, it also "derives its randomness from the exogenous variables", i.e., is defined on the same probability space. Each SCM implies a unique *observational distribution* over $\mathbf{V}$ (Pearl, 2000), but it also entails interventional distributions. Given a variable $A \in \mathbf{V}$, an intervention $A \leftarrow a$ amounts to replacing $f_A$ in $F$ with the constant function $A = a$. This yields a new SCM, which induces the *interventional distribution* under intervention $A \leftarrow a$. Similarly, we can intervene on multiple variables $\mathbf{V} \supseteq \mathbf{A} \leftarrow \mathbf{a}$. We then write $\mathbf{Y}_{\mathbf{a}}^*$ for the outcome in the intervened SCM, also called *potential outcome*. Note that the interventional distribution $\mathbb{P}_{\mathbf{Y}_{\mathbf{a}}^*}(\mathbf{y})$ differs in general from the conditional distribution $\mathbb{P}_{\mathbf{Y}|\mathbf{A}}(\mathbf{y} \mid \mathbf{a})$. This could for instance happen due to unobserved confounding effects.[1] We can also condition on a set of variables $\mathbf{W} \subseteq \mathbf{V}$ in the (observational distribution of the) original SCM before performing an intervention, which we denote by $\mathbb{P}_{\mathbf{Y}_{\mathbf{a}}^*|\mathbf{W}}(\mathbf{y} \mid \mathbf{w})$. This is a *counterfactual distribution*: "Given that we have observed $\mathbf{W} = \mathbf{w}$, what would $\mathbf{Y}$ have been had we set $\mathbf{A} \leftarrow \mathbf{a}$, instead of the value $\mathbf{A}$ had actually taken?" Note that the sets $\mathbf{A}$ and $\mathbf{W}$ need not be disjoint. We can now define counterfactual invariance.

**Definition 2.2** (Counterfactual invariance). *Let $\mathbf{A}$, $\mathbf{W}$ be (not necessarily disjoint) sets of nodes in a given SCM. A predictor $\hat{\mathbf{Y}}$ is counterfactually invariant in $\mathbf{A}$ with respect to $\mathbf{W}$, if $\mathbb{P}_{\hat{\mathbf{Y}}_{\mathbf{a}}^*|\mathbf{W}}(\mathbf{y} \mid \mathbf{w}) = \mathbb{P}_{\hat{\mathbf{Y}}_{\mathbf{a}'}^*|\mathbf{W}}(\mathbf{y} \mid \mathbf{w})$ almost surely, for all $\mathbf{a}, \mathbf{a}'$ in the domain of $\mathbf{A}$ and all $\mathbf{w}$ in the domain of $\mathbf{W}$.[2]*

A counterfactually invariant predictor can be viewed as robust to changes of $\mathbf{A}$, in the sense that the (conditional) post-interventional distribution of $\hat{\mathbf{Y}}$ does not change for different values of the intervention. Our Definition 2.2 is more general than previously considered notions of counterfactual invariance. For instance, the invariance in Definition 1.1 by Veitch et al. (2021) requires $\hat{\mathbf{Y}}_{\mathbf{a}}^* = \hat{\mathbf{Y}}_{\mathbf{a}'}^*$ almost surely for all $\mathbf{a}, \mathbf{a}'$ in the domain of $\mathbf{A}$. First, it does not allow to condition on observed evidence, i.e., it cannot consider "true counterfactuals" and is thus unable to promote—for

---

[1] We use $\mathbb{P}$ for distributions as is common in the kernel literature (Muandet et al., 2021) and the potential outcome notation $\mathbf{Y}_{\mathbf{a}}^*$ instead of $\mathbf{Y} \mid do(\mathbf{a})$ for conciseness when mixing conditioning with interventions.

[2] With a mild abuse of notation, if $\mathbf{W} = \emptyset$ then the requirement of conditional counterfactual invariance becomes $\mathbb{P}_{\hat{\mathbf{Y}}_{\mathbf{a}}^*}(\mathbf{y}) = \mathbb{P}_{\hat{\mathbf{Y}}_{\mathbf{a}'}^*}(\mathbf{y})$ almost surely, for all $\mathbf{a}, \mathbf{a}'$ in the domain of $\mathbf{A}$.

example—counterfactual fairness, see Definition 3.6 (Kusner et al., 2017). Second, it may appear stronger in that it asks for equality of random variables instead of equality of distributions. However, (a) contrary to their definition, Veitch et al. (2021) only enforce equality of distributions in practice (via MMD), and (b) since $\hat{\mathbf{Y}}_{\mathbf{a}}^*, \hat{\mathbf{Y}}_{\mathbf{a}'}^*$ are (deterministic) functions of the *same* exogenous (unobserved) random variables, distributional equality is a natural choice for counterfactual invariance. We remark that a notion of invariance has been studied by Mouli & Ribeiro (2022). In this work, the authors focus on learning classifiers that are counterfactually invariant to distribution shift using asymmetry learning. To the best of our knowledge, however, our work is the first attempt to provide a general graphical criterion for invariance, which can be verified from observational data.

**Kernel mean embeddings and conditional measures.** Our new objective function heavily relies on kernel mean embeddings (KMEs). We now highlight the main concepts pertaining KMEs and refer the reader to Smola et al. (2007); Schölkopf et al. (2002); Berlinet & Thomas-Agnan (2011); Muandet et al. (2017) for more details. Fix a measurable space $\mathscr{Y}$ with respect to a $\sigma$-algebra $\mathcal{F}_{\mathscr{Y}}$, and consider a probability measure $\mathbb{P}$ on the space $(\mathscr{Y}, \mathcal{F}_{\mathscr{Y}})$. Let $\mathcal{H}$ be a reproducing kernel Hilbert space (RKHS) with a bounded kernel $k_{\mathbf{Y}} : \mathscr{Y} \times \mathscr{Y} \to \mathbb{R}$, i.e., $k_{\mathbf{Y}}$ is such that $\sup_{\mathbf{y} \in \mathscr{Y}} k(\mathbf{y}, \mathbf{y}) < +\infty$. The kernel mean embedding $\mu_{\mathbb{P}}$ of $\mathbb{P}$ is defined as the expected value of the function $k(\,\cdot\,, \mathbf{y})$ with respect to $\mathbf{y}$, i.e., $\mu_{\mathbb{P}} := \mathbb{E}\left[k(\,\cdot\,, \mathbf{y})\right]$. The definition of KMEs can be extended to conditional distributions (Fukumizu et al., 2013; Grünewälder et al., 2012; Song et al., 2009; 2013). Consider two random variables $\mathbf{Y}, \mathbf{Z}$, and denote with $(\Omega_{\mathbf{Y}}, \mathcal{F}_{\mathbf{Y}})$ and $(\Omega_{\mathbf{Z}}, \mathcal{F}_{\mathbf{Z}})$ the respective measurable spaces. These random variables induce a probability measure $\mathbb{P}_{\mathbf{Y}, \mathbf{Z}}$ in the product space $\Omega_{\mathbf{Y}} \times \Omega_{\mathbf{Z}}$. Let $\mathcal{H}_{\mathbf{Y}}$ be a RKHS with a bounded kernel $k_{\mathbf{Y}}(\cdot, \cdot)$ on $\Omega_{\mathbf{Y}}$. We define the KME of a conditional distribution $\mathbb{P}_{\mathbf{Y}|\mathbf{Z}}(\cdot \mid \mathbf{z})$ via $\mu_{\mathbf{Y}|\mathbf{Z}=\mathbf{z}} := \mathbb{E}\left[k_{\mathbf{Y}}(\,\cdot\,, \mathbf{y}) \mid \mathbf{Z} = \mathbf{z}\right]$. Here, the expected value is taken over $\mathbf{y}$. KMEs of conditional measures can be estimated from samples. To illustrate this, consider i.i.d. samples $(\mathbf{y}_1, \mathbf{z}_1), \ldots, (\mathbf{y}_n, \mathbf{z}_n)$. Denote with $\hat{K}_{\mathbf{Y}}$ the kernel matrix with entries $[\hat{K}_{\mathbf{Y}}]_{i,j} := k_{\mathbf{Y}}(\mathbf{y}_i, \mathbf{y}_j)$. Furthermore, let $k_{\mathbf{Z}}$ be a bounded kernel on $\Omega_{\mathbf{Z}}$. Then, $\mu_{\mathbf{Y}|\mathbf{Z}=\mathbf{z}}$ can be estimated as

$$\hat{\mu}_{\mathbf{Y}|\mathbf{Z}=\mathbf{z}} := \sum_{i=1}^{n} \hat{w}_{\mathbf{Y}|\mathbf{Z}}^{(i)}(\mathbf{z}) k_{\mathbf{Y}}(\cdot, \mathbf{y}_i), \quad \hat{w}_{\mathbf{Y}|\mathbf{Z}}(\cdot) := (\hat{K}_{\mathbf{Z}} - n\lambda I)^{-1} \left[k_{\mathbf{Z}}(\cdot, \mathbf{z}_1), \cdots, k_{\mathbf{Z}}(\cdot, \mathbf{z}_n)\right]^T \quad (1)$$

where, $I$ is the identity matrix and $\lambda$ is a regularization parameter. Here, $\hat{w}_{\mathbf{X}|\mathbf{A}}^{(i)}(\cdot)$, the $i$-th entry of $\hat{w}_{\mathbf{X}|\mathbf{A}}(\cdot)$, are the coefficients of kernel ridge regression (Grünewälder et al., 2012).

## 3 COUNTERFACTUALLY INVARIANT PREDICTORS

We now establish a simple graphical criterion to express counterfactual invariance as a conditional independence in the observational distribution, rendering it estimable from i.i.d. data. We first repeat the notion of *blocked paths* (Pearl, 2000).

**Definition 3.1.** *Consider a path $\pi$ of causal graph $\mathcal{G}$. A set of nodes $\mathbf{Z}$ blocks $\pi$, if $\pi$ contains a triple of consecutive nodes connected in one of the following ways: $N_i \to Z \to N_j$, $N_i \leftarrow Z \to N_j$, or $N_i \to M \leftarrow N_j$, with $N_i, N_j \notin \mathbf{Z}$, $Z \in \mathbf{Z}$, and neither $M$ nor any descendent of $M$ is in $\mathbf{Z}$.*

**Theorem 3.2.** *Let $\mathcal{G}$ be a causal diagram, and let $\mathbf{A}$, $\mathbf{W}$ be two (not necessarily disjoint) sets of nodes in $\mathcal{G}$. Let $\mathbf{Z}$ be a set of nodes that blocks all non-causal[3] paths from $\mathbf{A} \cup \mathbf{W}$ to $\mathbf{Y}$. Then, for any SCM compatible with $\mathcal{G}$, any predictor $\hat{\mathbf{Y}}$ that satisfies $\hat{\mathbf{Y}} \perp\!\!\!\perp \mathbf{A} \mid \mathbf{Z}$ is counterfactually invariant in $\mathbf{A}$ with respect to $\mathbf{W}$.*

The proof is deferred to Appendix A. Our key observation is that the valid set $\mathbf{Z}$ as in Theorem 3.2 acts as a $d$-separator for certain random variables in a graph that allows reasoning about dependencies among pre- and post-interventional random variables. This graph simplifies the counterfactual graph by Shpitser & Pearl (2008), and it generalizes the augmented graph structure described in Theorem 1 by Shpitser & Pearl (2009). We can then combine the Markov property with covariate adjustment to prove the claim. However, our proof does *not* rely on identification of the counterfactual distributions (e.g., by simply applying the do-calculus (Pearl, 2000)). Crucially, *the existence of a measurable set $\mathbf{Z}$ as in Theorem 3.2 does not imply the identifiability of counterfactual distributions $\mathbb{P}_{\mathbf{Y}_{\mathbf{a}}^*}(\mathbf{y})$* (see Figure 1(a) for a counterexample). In particular, the assumptions do not rule out

---

[3]A non-causal paths from $\mathbf{A} \cup \mathbf{W}$ to $\mathbf{Y}$ is a path connecting $\mathbf{A} \cup \mathbf{W}$ and $\mathbf{Y}$ in which at least one edge points against causal ordering. Shpitser et al. (2012)

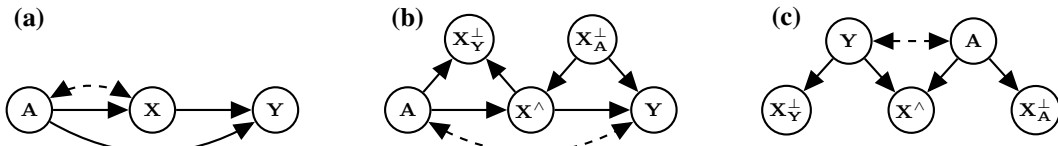

Figure 1: **(a) An example for Theorem 3.2**, where $\mathbb{P}_{\mathbf{Y}_{\mathbf{a}}^*}(\mathbf{y})$ is unidentifiable, due to the confounding effect denoted by the dashed double arrow. However, by Theorem 3.2 any predictor $\hat{\mathbf{Y}}$ such that $\hat{\mathbf{Y}} \perp\!\!\!\perp \mathbf{A} \mid \mathbf{Z}$ with $\mathbf{Z} = \{\mathbf{X}\}$ is counterfactually invariant in $\mathbf{A}$. Hence, the criterion in Theorem 3.2 is weaker than identifiability. **(b)-(c) Causal and anti-causal structure** as in Veitch et al. (2021). The variable $\mathbf{X}$ is decomposed in three parts. $\mathbf{X}_{\mathbf{A}}^{\perp}$ the part of $\mathbf{X}$ that is not causally influenced by $\mathbf{A}$, $\mathbf{X}_{\mathbf{Y}}^{\perp}$ is the part that does not causally influence $\mathbf{Y}$, and $\mathbf{X}^{\wedge}$ is the remaining part, that is both influenced by $\mathbf{A}$ and that influences $\mathbf{Y}$.

hidden confounding in the model. Hence, our method is applicable even when (certain parameters of) $\mathbb{P}_{\mathbf{Y}_{\mathbf{a}}^*}(\mathbf{y})$ or $\mathbb{P}_{\mathbf{Y}_{\mathbf{a}}^*|\mathbf{W}}(\mathbf{y})$ cannot be learned from observational data.

**Counterfactually invariant predictors.** We propose to use the HSCIC$(\hat{\mathbf{Y}}, \mathbf{A} \mid \mathbf{Z})$, to promote counterfactual invariance by encouraging the sufficient conditional independence $\hat{\mathbf{Y}} \perp\!\!\!\perp \mathbf{A} \mid \mathbf{Z}$. Given a loss function $\mathcal{L}(\hat{\mathbf{Y}})$, we propose to minimize the following loss

$$\mathcal{L}_{\text{TOTAL}}(\hat{\mathbf{Y}}) = \mathcal{L}(\hat{\mathbf{Y}}) + \gamma \cdot \text{HSCIC}(\hat{\mathbf{Y}}, \mathbf{A} \mid \mathbf{Z}) \,, \tag{2}$$

with a parameter $\gamma \geq 0$, regulating the trade-off between accuracy and counterfactual invariance. There is no principled way to choose the "right" trade-off parameter. However, in practice we can use a standard approach, which consists of two steps: (i) learn a collection of models on the Pareto frontier using different values of $\gamma$; (ii) from a collection of models choose the best one following, i.e., Muandet et al. (2021). Next, we develop and justify using the HSCIC, (see Corollary 3.5) for which HSCIC$(\hat{\mathbf{Y}}, \mathbf{A} \mid \mathbf{Z}) = 0$ if and only if $\hat{\mathbf{Y}} \perp\!\!\!\perp \mathbf{A} \mid \mathbf{Z}$.

**Hilbert-Schmidt Conditional Independence Criterion (HSCIC).** Consider two random variables $\mathbf{Y}$ and $\mathbf{A}$, and denote with $(\Omega_{\mathbf{Y}}, \mathcal{F}_{\mathbf{Y}})$ and $(\Omega_{\mathbf{A}}, \mathcal{F}_{\mathbf{A}})$ the respective measurable spaces. Suppose that we are given two RKHSs $\mathcal{H}_{\mathbf{Y}}, \mathcal{H}_{\mathbf{A}}$ over the support of $\mathbf{Y}$ and $\mathbf{A}$ respectively. The tensor product space $\mathcal{H}_{\mathbf{Y}} \otimes \mathcal{H}_{\mathbf{A}}$ is defined as the space of functions of the form $(f \otimes g)(\mathbf{y}, \mathbf{a}) \coloneqq f(\mathbf{y})g(\mathbf{a})$, for all $f \in \mathcal{H}_{\mathbf{Y}}$ and $g \in \mathcal{H}_{\mathbf{A}}$. The tensor product space yields a natural RKHS structure, with kernel $k$ defined by $k(\mathbf{y} \otimes \mathbf{a}, \mathbf{y}' \otimes \mathbf{a}') \coloneqq k_{\mathbf{Y}}(\mathbf{y}, \mathbf{y}')k_{\mathbf{A}}(\mathbf{a}, \mathbf{a}')$. We refer the reader, i.e., to Szabó & Sriperumbudur (2017) for more details on tensor product spaces. With this notation we define:

**Definition 3.3** (HSCIC, following Definition 5.3 by Park & Muandet (2020))**.** *For (sets of) random variables* $\mathbf{Y}$, $\mathbf{A}$, $\mathbf{Z}$ *the* HSCIC *between* $\mathbf{Y}$ *and* $\mathbf{A}$ *given* $\mathbf{Z}$ *is defined as the real-valued random variable* HSCIC$(\mathbf{Y}, \mathbf{A} \mid \mathbf{Z}) = H_{\mathbf{Y},\mathbf{A}|\mathbf{Z}} \circ \mathbf{Z}$*. Here,* $H_{\mathbf{Y},\mathbf{A}|\mathbf{Z}}$ *is a real-valued deterministic function, defined as* $H_{\mathbf{Y},\mathbf{A}|\mathbf{Z}}(\mathbf{z}) \coloneqq \left\| \mu_{\mathbf{Y},\mathbf{A}|\mathbf{Z}=\mathbf{z}} - \mu_{\mathbf{Y}|\mathbf{Z}=\mathbf{z}} \otimes \mu_{\mathbf{A}|\mathbf{Z}=\mathbf{z}} \right\|$*, with* $\|\cdot\|$ *the norm induced by the inner product of the tensor product space* $\mathcal{H}_{\mathbf{X}} \otimes \mathcal{H}_{\mathbf{A}}$*.*

Our Definition 3.3 differs slightly from Park & Muandet (2020), who define HSCIC using the Bochner conditional expected value. While it is functionally equivalent (with the same implementation, see eq. (3)), it has the benefit of bypassing some technical assumptions required by Park & Muandet (2020). We refer readers to Appendix C and D for a comparison with previous approaches. The HSCIC has the following important property.

**Theorem 3.4** (following Theorem 5.4 by Park & Muandet (2020))**.** *If the kernel* $k$ *of* $\mathcal{H}_{\mathbf{X}} \otimes \mathcal{H}_{\mathbf{A}}$ *is characteristic*[4]*,* HSCIC$(\mathbf{Y}, \mathbf{A} \mid \mathbf{Z}) = 0$ *almost surely if and only if* $\mathbf{Y} \perp\!\!\!\perp \mathbf{A} \mid \mathbf{Z}$*.*

A proof is in Appendix B. We remark that "most interesting" kernels such as the Gaussian and Laplacian kernel are characteristic. Furthermore, if kernels are translation-invariant and characteristic, then their tensor product is also a characteristic kernel (Szabó & Sriperumbudur, 2017). Hence, this natural assumption is non-restrictive in practice. Combining Theorems 3.2 and 3.4, we can now use HSCIC to promote counterfactual invariance.

**Corollary 3.5.** *Consider an SCM with causal diagram* $\mathcal{G}$ *and fix two (not necessarily disjoint) sets of nodes* $\mathbf{A}$, $\mathbf{W}$*. Let* $\mathbf{Z}$ *be a set of nodes in* $\mathcal{G}$ *that blocks all* non-causal *paths from* $\mathbf{A} \cup \mathbf{W}$ *to*

---

[4]The tensor product kernel $k$ is characteristic if the mapping $\mathbb{P}_{\mathbf{Y},\mathbf{A}} \mapsto \mathbb{E}_{\mathbf{y},\mathbf{a}}\left[k(\,\cdot\,, \mathbf{y} \otimes \mathbf{a})\right]$ is injective.

**Y**. *Then, any predictor* $\hat{\mathbf{Y}}$ *that satisfies* $\mathrm{HSCIC}(\hat{\mathbf{Y}}, \mathbf{A} \mid \mathbf{Z}) = 0$ *almost surely is counterfactually invariant in* $\mathbf{A}$ *with respect to* $\mathbf{W}$.

We do *not* require $\mathbf{A}$, $\mathbf{W}$, $\mathbf{Z}$ or $\mathbf{Y}$ to be binary or categorical, a major improvement over existing methods (Chiappa, 2019; Xu et al., 2020), which cannot handle continuous multi-variate attributes.

**Estimating the HSCIC from samples.** Given $n$ samples $\{(\mathbf{y}_i, \mathbf{a}_i, \mathbf{z}_i)\}_{i=1}^n$, denote by $\hat{K}_{\hat{\mathbf{Y}}}$ and $\hat{K}_{\mathbf{A}}$ the corresponding kernel matrices for $\hat{\mathbf{Y}}$ and $\mathbf{A}$ (see eq. (1)). We then estimate the $H_{\hat{\mathbf{Y}}, \mathbf{A} \mid \mathbf{X}}$ as

$$\hat{H}^2_{\hat{\mathbf{Y}}, \mathbf{A} \mid \mathbf{Z}}(\cdot) = \hat{w}^T_{\hat{\mathbf{Y}}, \mathbf{A} \mid \mathbf{Z}}(\cdot) \left( \hat{K}_{\hat{\mathbf{Y}}} \odot \hat{K}_{\mathbf{A}} \right) \hat{w}_{\hat{\mathbf{Y}}, \mathbf{A} \mid \mathbf{Z}}(\cdot) - 2 \left( \hat{w}^T_{\hat{\mathbf{Y}} \mid \mathbf{Z}}(\cdot) \hat{K}_{\mathbf{Y}} \hat{w}_{\hat{\mathbf{Y}}, \mathbf{A} \mid \mathbf{X}}(\cdot) \right) \tag{3}$$
$$\cdot \left( \hat{w}^T_{\mathbf{A} \mid \mathbf{X}}(\cdot) \hat{K}_{\mathbf{A}} \hat{w}_{\hat{\mathbf{Y}}, \mathbf{A} \mid \mathbf{Z}}(\cdot) \right) + \left( \hat{w}^T_{\hat{\mathbf{Y}} \mid \mathbf{Z}}(\cdot) \hat{K}_{\hat{\mathbf{Y}}} \hat{w}_{\hat{\mathbf{Y}} \mid \mathbf{Z}}(\cdot) \right) \left( \hat{w}^T_{\mathbf{A} \mid \mathbf{Z}}(\cdot) \hat{K}_{\mathbf{A}} \hat{w}_{\mathbf{A} \mid \mathbf{Z}}(\cdot) \right) \,,$$

where $\odot$ is element-wise multiplication. The vectors $\hat{w}_{\hat{\mathbf{Y}} \mid \mathbf{Z}}(\cdot)$, $\hat{w}_{\mathbf{A} \mid \mathbf{Z}}(\cdot)$, and $\hat{w}_{\hat{\mathbf{Y}}, \mathbf{A} \mid \mathbf{Z}}(\cdot)$ are found via kernel ridge regression. Caponnetto & Vito (2007) the convergence (and rates) of $\hat{H}^2_{\hat{\mathbf{Y}}, \mathbf{A} \mid \mathbf{Z}}(\cdot)$ to $H^2_{\hat{\mathbf{Y}}, \mathbf{A} \mid \mathbf{Z}}(\cdot)$ under mild conditions. In practice, computing the HSCIC approximation $\hat{H}^2_{\hat{\mathbf{Y}}, \mathbf{A} \mid \mathbf{Z}}(\cdot)$ may be time-consuming. To speed it up, we can use random Fourier features to approximate the matrices $\hat{K}_{\hat{\mathbf{Y}}}$ and $\hat{K}_{\mathbf{A}}$ (Rahimi & Recht, 2007; Avron et al., 2017). We emphasize that eq. (3) allows us to consistently estimate the HSCIC *from observational i.i.d. samples, without prior knowledge of the counterfactual distributions.*

**Measuring counterfactual invariance.** Besides predictive performance, e.g., mean squared error (**MSE**) for regression, our key metric of interest is the level of counterfactual invariance achieved by the predictor $\hat{\mathbf{Y}}$, i.e., a measure of how the distribution of $\hat{\mathbf{Y}}_{\mathbf{a}}^*$ changes for different values of $\mathbf{a}$ and across all conditioning values $\mathbf{w}$. We quantify the overall counterfactual variance as a single scalar, the **V**ariance of **C**ounter**F**actuals (**VCF**):

$$\mathbf{VCF}(\hat{\mathbf{Y}}) = \mathbb{E}_{\mathbf{w} \sim \mathbb{P}_{\mathbf{W}}} \left[ \mathrm{var}_{\mathbf{a}' \sim \mathbb{P}_{\mathbf{A}}} [\mathbb{E}_{\hat{\mathbf{Y}}_{\mathbf{a}'}^* \mid \mathbf{W} = \mathbf{w}}(\hat{\mathbf{y}} \mid \mathbf{w})] \right] \,. \tag{4}$$

That is, we look at how the average outcome varies with the interventional value $\mathbf{a}$ at conditioning value $\mathbf{w}$ and average this variance over $\mathbf{w}$. For deterministic predictors (i.e., point estimators), which we use in all our experiments, the variance term in eq. (4) is zero if and only if counterfactual invariance holds at $\mathbf{w}$ for (almost) all $\mathbf{a}'$ in the support of $\mathbb{P}_{\mathbf{A}}$ (i.e., the prediction remains constant). Since the variance is non-negative, **VCF** (we often drop the argument) is then zero if and only if counterfactual invariance holds almost surely. To estimate **VCF** in practice, we pick $d$ datapoints $(\mathbf{w}_i)_{i=1}^d$ from the observed data, and for each compute the counterfactual outcomes $\hat{\mathbf{Y}}_{\mathbf{a}'}^* \mid \mathbf{w}_i$ for $k$ different values of $\mathbf{a}'$. The inner expectation is simply the predictor output. We use empirical variances with $k$ examples for each of the $d$ chosen datapoints, and the empirical mean of the $d$ variances for the outer expectation. Crucially, **VCF** requires access to ground-truth counterfactual distributions, which by their very nature are unavailable in practice (neither for training nor at test time). Hence, we can only assess **VCF**, as a direct measure of counterfactual invariance, in synthetic scenarios. We demonstrate in our experiments, that HSCIC (estimable from the observed data) empirically serves as a proxy for **VCF**.

**Applications of counterfactual invariance.** We briefly outline potential applications of counterfactual invariance, which we will subsequently study empirically on real-world data.

*Image classification.* Counterfactual invariance serves as a strong notion of robustness in high-dimensional settings, such as image classification: "Would the truck have been classified correctly had it been winter in this exact situation instead of summer?" For concrete demonstration, we will use the dSprites dataset (Matthey et al., 2017), which consists of relatively simple, yet high-dimensional, square black and white images of different shapes (squares, ellipses, etc.), sizes, and orientations (rotation) in different xy-positions.

*Counterfactual fairness.* The popular definition of counterfactual fairness (Kusner et al., 2017) is an incarnation of our counterfactual invariance (see Definition 2.2). It is captured informally by the following question after receiving a consequential decision: "Would I have gotten the same outcome had I been a different gender, race, or age with all else being equal?". Again, we denote the outcome by $\mathbf{Y} \subset \mathbf{V}$ and so-called *protected attributes* by $\mathbf{A} \subseteq \mathbf{V} \setminus \mathbf{Y}$ such as gender, race, or age protected under anti-discrimination laws (see, e.g., Barocas & Selbst (2016)). Collecting all

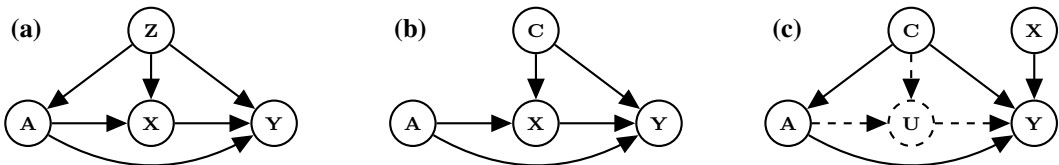

Figure 2: **(a) Assumed causal structure for the synthetic experiments** (see Section 4.1). The precise corresponding generative random variables are described in Appendix E. **(b) Assumed causal structure for the Adult dataset**, where $\mathbf{A}$ consists of the protected attributes *gender* and *age*.**(c) Causal structure for the constructed dSprites ground truth**, where $\mathbf{A} = \{\text{Pos.X}\}$, $\mathbf{U} = \{\text{Scale}\}$, $\mathbf{C} = \{\text{Shape, Pos.Y}\}$, $\mathbf{X} = \{\text{Color, Orientation}\}$, and $\mathbf{Y} = \{\text{Outcome}\}$. $\mathbf{U}$ is unobserved.

remaining observed covariates into $\mathbf{W} \coloneqq \mathbf{V} \setminus \mathbf{Y}$, the definition of counterfactual fairness clearly becomes a special case of our counterfactual invariance.

**Definition 3.6** (Counterfactual Fairness, Definition 5 by Kusner et al. (2017)). *A predictor $\hat{\mathbf{Y}}$ is* counterfactually fair *if under any context* $\mathbf{W} = \mathbf{w}$ *and* $\mathbf{A} = \mathbf{a}$*, it holds* $\mathbb{P}_{\hat{\mathbf{Y}}^*_{\mathbf{a}}|\mathbf{W},\mathbf{A}}(\mathbf{y} \mid \mathbf{w},\mathbf{a}) = \mathbb{P}_{\hat{\mathbf{Y}}^*_{\mathbf{a}'}|\mathbf{W},\mathbf{A}}(\mathbf{y} \mid \mathbf{w},\mathbf{a})$*, for all* $\mathbf{y}$ *and for any value* $\mathbf{a}'$ *attainable by* $\mathbf{A}$.

*Text classification.* Veitch et al. (2021) motivate the importance of counterfactual invariance in text classification tasks. We now provide a detailed comparison with our method on the (limited) causal models studied by Veitch et al. (2021), emphasizing that our definition and method applies to a much broader class of causal models. Specifically, we consider the causal and anti-causal structures (Figure 1 in Veitch et al. (2021)). Both diagrams consist of a protected attribute $\mathbf{Z}$, an observed covariate $\mathbf{X}$, and the outcome $\mathbf{Y}$. Veitch et al. (2021) provide *necessary* conditions for counterfactual invariance. They prove that if $\hat{\mathbf{Y}}^*_{\mathbf{z}} = \hat{\mathbf{Y}}^*_{\mathbf{z}'}$ almost surely, then *assuming their proposed causal and anti-causal structures* (see Figure 1(b-c)) it holds $\hat{\mathbf{Y}} \perp\!\!\!\perp \mathbf{Z}$ and $\hat{\mathbf{Y}} \perp\!\!\!\perp \mathbf{Z} \mid \mathbf{Y}$, respectively. That is, in practice they enforce a consequence of the desired criterion instead of a prerequisite. Our work complements the results by Veitch et al. (2021), as shown in the following corollary, which is a direct consequence of Theorem 3.2.

**Corollary 3.7.** *Under the causal and anti-causal graph, suppose that $\mathbf{Z}$ and $\mathbf{Y}$ are not confounded. If $\hat{\mathbf{Y}} \perp\!\!\!\perp \mathbf{Z}$, it holds $\mathbb{P}_{\hat{\mathbf{Y}}^*_{\mathbf{z}}}(\mathbf{y}) = \mathbb{P}_{\hat{\mathbf{Y}}^*_{\mathbf{z}'}}(\mathbf{y})$ almost surely, for all $\mathbf{z}, \mathbf{z}'$ in the domain of $\mathbf{Z}$.*

We remark that the notion of invariance studied by Veitch et al. (2021) is enforced on the true counterfactuals. Our definition, and Corollary 3.7, on the other hand, only requires invariance of the resulting distribution, which is a weaker requirement than Veitch et al. (2021). However, we provide experiments in Appendix E.6 to show that our method, based on Corollary 3.7, and the method by Veitch et al. (2021) have a similar effect in practice.

## 4 EXPERIMENTS

### 4.1 SYNTHETIC EXPERIMENTS

We begin our empirical assessment of HSCIC on synthetic datasets, where ground truth is known and systematically study how performance is influenced by the dimensionality of the variables. We simulate different datasets following to the causal graphical structure in Figure 2(a). The datasets are composed of four sets of observed continuous variables: (i) the prediction target $\mathbf{Y}$, (ii) the variable(s) we want to be counterfactually invariant in $\mathbf{A}$, (iii) observed covariates that mediate effects from $\mathbf{A}$ on $\mathbf{Y}$, and (iv) observed confounding variables $\mathbf{Z}$. The goal is to learn a predictor $\hat{\mathbf{Y}}$ that is counterfactually invariant in $\mathbf{A}$ with respect to $\mathbf{W} \coloneqq \mathbf{A} \cup \mathbf{X} \cup \mathbf{Z}$. We consider various artificially generated datasets for this example, which mainly differ in the dimension of the observed variables and their correlations. A detailed description of each dataset is deferred to Appendix E.

**Model choices and parameters.** For all synthetic experiments, we train the model using fully connected networks (MLPs). We use the **MSE** loss $\mathcal{L}_{\text{MSE}}(\hat{\mathbf{Y}})$ as the predictive loss $\mathcal{L}$ in eq. (2) for continuous outcomes $\mathbf{Y}$. We generate 4k samples from the observational distribution in each setting and use an 80 to 20 train to test split. All metrics reported are on the test set. We perform hyperparameter tuning for MLP hyperparameters based on a random strategy (see Appendix E for details). The HSCIC($\hat{\mathbf{Y}}, \mathbf{A} \mid \mathbf{Z}$) term is computed as in eq. (3) using a Gaussian kernel with amplitude 1.0 and length scale 0.1. The regularization parameter $\lambda$ for the ridge regression coefficients in eq. (1) is set to $\lambda = 0.01$. We set $d = 1000$ and $k = 500$ in the estimation of **VCF**.

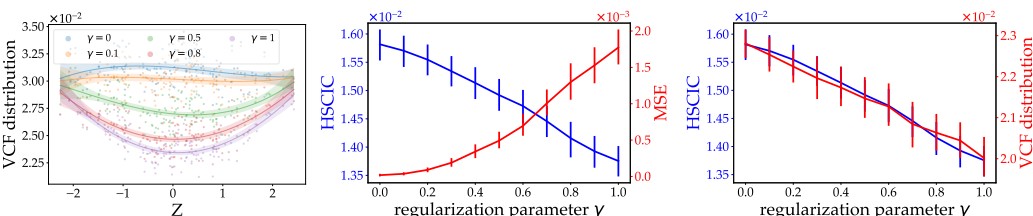

Figure 3: **(Left) variance of the counterfactual distributions** for 100 random datapoints with lines representing $3^{\text{rd}}$-order polynomial regression and shaded areas being 95% confidence intervals. **(Center) trade-off between the accuracy and counterfactual invariance.** We observe that the HSCIC decreases, as the **MSE** increases. Vertical bars denote standard errors over 15 different random seeds. **(Right) Correspondence between the HSCIC and the VCF**, for increasing $\gamma$. Again, vertical bars denote standard errors over 15 different random seeds.

**Model performance.** We first perform a set of experiments to study the effect of the HSCIC, and to highlight the trade-off between accuracy and counterfactual invariance. For this set of experiments, we generate a dataset as described in Appendix E.1. Figure 3 (left) shows the variance term of **VCF** for different regularization parameters $\gamma$, as a function of the values of $\mathbf{Z}$ (i.e., before taking the outer expectation). We observe that increasing values of $\gamma$ lead to a consistent decrease of the variances w.r.t. the interventional value $\mathbf{a}$. Figure 3 (center) shows the values attained by the HSCIC and **MSE** for increasing $\gamma$, demonstrating the expected trade-off in raw predictive performance and enforcing counterfactual invariance. Finally, Figure 3 (right) highlights the usefulness of HSCIC as a measure of counterfactual invariance, being in strong agreement with **VCF** (see discussion after eq. (4)).

**Comparison with baselines.** We now compare our method against baselines in two settings, which we refer to as Scenario 1 and Scenario 2. These two settings differ in how the conditioning set $\mathbf{Z}$ affects the outcome $\mathbf{Y}$, with Scenario 1 exhibiting higher correlation of $\mathbf{Z}$ with both the mediator $\mathbf{X}$ and the outcome $\mathbf{Y}$ (see Appendix E.2 for details). Since counterfactually invariant training has not received much attention yet, our our choice of baselines for experimental comparison is highly limited. Corollary 3.7 together with the fact that Veitch et al. (2021) in practice only enforce distributional equality implies that our method subsumes theirs in the causal setting they have proposed. We benchmarked our method against Veitch et al. (2021) in the limited causal and anti-causal settings of Figure 6(b-c) in Appendix E.6, showing that our method performs on par or better than theirs. Since counterfactual fairness is a special case of counterfactual invariance, we also compare against two methods proposed by Kusner et al. (2017) (in applicable settings). We compare to the *Level 1* (only use non-descendants of $\mathbf{A}$ as inputs to $\hat{\mathbf{Y}}$) and the *Level 3* (assume an additive noise model and in addition to non-descendants, only use the residuals of descendants of $\mathbf{A}$ after regression on $\mathbf{A}$ as inputs to $\hat{\mathbf{Y}}$) approaches of Kusner et al. (2017). We refer to these two baselines as CF1 and CF2 respectively. We summarize the results on the two scenarios in Table 1. For a suitable choice of $\gamma$, our method outperforms the baseline CF2 in both **MSE** and **VCF** simultaneously. While CF1 satisfies counterfactual invariance perfectly by construction (**VCF** $= 0$), its **MSE** is substantially higher than for all other methods. Our method provides to flexibly trade predictive performance for counterfactual invariance via a single tuning knob $\lambda$ and pareto-dominates existing methods.

**Multi-dimensional variables.** We perform a third set of experiments to assess HSCIC's performance in higher dimensions. We consider simulated datasets (described in Appendix E.3), where we independently increase the dimension of $\mathbf{Z}$ and $\mathbf{A}$ in two different simulations, leaving the rest of the variables unchanged. The results in Tables 2 for different regularization coefficients $\gamma$ and different dimensions of $\mathbf{A}$ and $\mathbf{Z}$ demonstrate that HSCIC can handle multi-dimensional variables while maintaining performance, as counterfactual invariance is approached when $\gamma$ increases. We provide results where both $\mathbf{A}$ and $\mathbf{Z}$ are multivariate in Appendix E.3.

### 4.2 HIGH-DIMENSIONAL IMAGE EXPERIMENTS

We consider the image classification task on the dSprites dataset (Matthey et al., 2017). Since this dataset is fully synthetic and labelled (we know all factors for each image), we consider a causal model as depicted in Figure 2(c). The full structural equations are provided in Appendix E.4, where we assume a causal graph over the determining factors of the image, and essentially look up the corresponding image in the simulated dataset. This experiment is particularly challenging due to the mixed categorical and continuous variables in $\mathbf{C}$ (`shape`, `y-pos`) and $\mathbf{X}$ (`color`, `orientation`), continuous $\mathbf{A}$ (`x-pos`). All variables except for scale are assumed to be ob-

Table 1: **Performance of the HSCIC against baselines** CF1 and CF2 on two synthetic datasets (see Appendix E.2). Notably, for $\gamma = 5$ in Scenario 1 and $\gamma = 13$ in Scenario 2 we outperform CF2 in **MSE** and **VCF** simultaneously.

| | Scenario 1 | | | Scenario 2 | | |
|---|---|---|---|---|---|---|
| | **MSE** $\times 10^3$ | **HSCIC** $\times 10^5$ | **VCF** $\times 10^3$ | **MSE** $\times 10^3$ | **HSCIC** $\times 10^5$ | **VCF** $\times 10^3$ |
| $\gamma = 0$ | $0.36 \pm 0.50$ | $1600 \pm 6$ | $20.33 \pm 1.00$ | $2.00 \pm 0.10$ | $1700 \pm 20$ | $239.0 \pm 0.4$ |
| $\gamma = 5$ | $17.00 \pm 0.03$ | $877 \pm 3$ | $6.84 \pm 1.00$ | $43.00 \pm 8.00$ | $1410 \pm 80$ | $230.0 \pm 6.0$ |
| $\gamma = 10$ | $19.80 \pm 2.00$ | $816 \pm 4$ | $5.89 \pm 0.60$ | $50.00 \pm 8.00$ | $1210 \pm 50$ | $190.0 \pm 3.0$ |
| $\gamma = 13$ | $22.00 \pm 1.00$ | $790 \pm 8$ | $5.78 \pm 0.50$ | $133.00 \pm 6.00$ | $990 \pm 30$ | $157.0 \pm 7.0$ |
| CF1 | $24.44 \pm 3.00$ | $790 \pm 5$ | $0.00$ | $218.00 \pm 10.00$ | $80 \pm 5$ | $0.0$ |
| CF2 | $19.50 \pm 2.00$ | $1400 \pm 10$ | $7.50 \pm 1.00$ | $137.00 \pm 4.00$ | $1770 \pm 30$ | $167.0 \pm 2.0$ |

Table 2: **Results of the MSE ($\times 10^5$ for readability), HSCIC, VCF for increasing dimension of A (top) and Z (bottom)**, on synthetic datasets as in Appendix E.3. All other variables are one-dimensional in both cases.

| | dimA=2 | | | dimA=5 | | | dimA=10 | | | dimA=20 | | |
|---|---|---|---|---|---|---|---|---|---|---|---|---|
| $\gamma$ | MSE | HSCIC | VCF | MSE | HSCIC | VCF | MSE | HSCIC | VCF | MSE | HSCIC | VCF |
| 0 | 35 | 0.0230 | 0.025 | 1.53 | 0.0177 | 0.013 | 10 | 0.0150 | 0.004 | 1.45 | 0.0126 | 0.0010 |
| $\frac{1}{2}$ | 86 | 0.0214 | 0.022 | 1580 | 0.0157 | 0.010 | 40 | 0.0135 | 0.002 | 33 | 0.0115 | 0.0004 |
| 1 | 210 | 0.0200 | 0.022 | 1450 | 0.0144 | 0.008 | 100 | 0.0127 | 0.001 | 60 | 0.0111 | 0.0003 |

| | dimZ=2 | | | dimZ=3 | | | dimZ=5 | | |
|---|---|---|---|---|---|---|---|---|---|
| $\gamma$ | MSE | HSCIC | VCF | MSE | HSCIC | VCF | MSE | HSCIC | VCF |
| 0 | 20 | 0.002024 | $6.15 \times 10^{-5}$ | 11 | 0.03407 | 0.023 | 53.9 | 0.033180 | 0.0062 |
| $\frac{1}{2}$ | 20 | 0.002023 | $1.29 \times 10^{-5}$ | 49 | 0.03406 | 0.016 | 21.9 | 0.033178 | 0.0061 |
| 1 | 20 | 0.002019 | $4.40 \times 10^{-6}$ | 57 | 0.03403 | 0.009 | 18 | 0.033176 | 0.0039 |

served, and all variables jointly with the actual high-dimensional image determine the outcome $\mathbf{Y}$. Our goal is to learn a predictor $\hat{\mathbf{Y}}$ that is counterfactually invariant in the x-position with respect to all other observed variables. In the chosen causal structure, $\{\texttt{shape}, \texttt{y-pos}\} \in \mathbf{C}$ block all non-causal paths from $\mathbf{A} \cup \mathbf{C} \cup \mathbf{U}$ to $\mathbf{Y}$. Hence, we seek to achieve $\hat{\mathbf{Y}} \perp\!\!\!\perp \texttt{x-pos} \mid \{\texttt{shape}, \texttt{y-pos}\}$ via the HSCIC operator. To accommodate the mixed input types, $\hat{\mathbf{Y}}$ puts an MLP on top of features extracted from the images via convolutional layers concatenated with features extracted from the remaining inputs via an MLP. Figure 4 demonstrates that HSCIC achieves improved **VCF** as $\gamma$ increases up to a certain point while affecting **MSE**, an inevitable trade-off.

### 4.3 FAIRNESS WITH CONTINUOUS PROTECTED ATTRIBUTES

We apply our method to the popular UCI Adult dataset (Kohavi & Becker, 1996). Our goal is to predict whether an individual's income is above a certain threshold $\mathbf{Y} \in \{0, 1\}$ based on a collection of (demographic) information including protected attributes such as gender and age. We follow Nabi & Shpitser (2018); Chiappa (2019), where a subset of variables are selected from the dataset and a causal structure is assumed as in Figure 2(b) (see Appendix E.5 and Figure 7 for details). We choose gender (considered binary in this dataset) and age (considered continuous) as the protected attributes $\mathbf{A}$. We denote the marital status, level of education, occupation, working hours per week, and work class jointly by $\mathbf{X}$ and combine the remaining observed attributes in $\mathbf{C}$. Our goal is to learn a predictor $\hat{\mathbf{Y}}$ that is counterfactually invariant in $\mathbf{A}$ with respect to $\mathbf{W} = \mathbf{C} \cup \mathbf{X}$. We remark that achieving fairness for continuous or even mixed categorical and continuous protected attributes is an ongoing line of research (even for non-causal fairness notions) (Mary et al., 2019; Chiappa & Pacchiano, 2021), but directly supported by HSCIC.

We use an MLP with binary cross-entropy loss for $\hat{\mathbf{Y}}$. Since this experiment is based on real data, the true counterfactual distribution cannot be known. Hence, we follow Chiappa & Pacchiano (2021) and estimate a possible true SCM by inferring the posterior distribution over the unobserved vari-

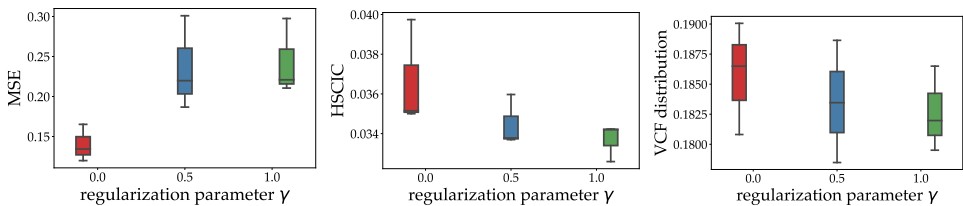

Figure 4: **Results of MSE, HSCIC operator and VCF for the dSprites image dataset experiment.** The HSCIC operator decreases steadily with higher values of $\gamma$. Similarly, a necessary increase of **MSE** can be observed. For both $\gamma = 0.5$ and $\gamma = 1$ an overall decrease of **VCF** is observed compared to the not-regularized setting.

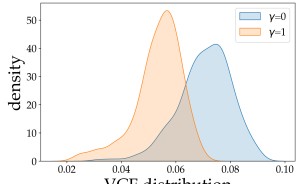

|  | Accuracy | HSCIC | VCF |
|---|---|---|---|
| $\gamma = 0$ | $(81.725 \pm 0.0481)\%$ | $0.01067 \pm 3.4 \times 10^{-5}$ | $0.06685 \pm 0.006$ |
| $\gamma = 1$ | $(81.547 \pm 0.0745)\%$ | $0.00739 \pm 6.8 \times 10^{-5}$ | $0.05156 \pm 0.003$ |

Figure 5: **(Left) Distribution of VCF values** (unnormalized) for different choices of $\gamma$. We observe less variance and more mass near zero for regularization parameter $\gamma = 1$. **(Right) Results on accuracy, HSCIC and VCF**, showing a considerable improvement of HSCIC and **VCF** for $\gamma = 1$.

ables using variational autoencoders (Kingma & Welling, 2014). Even though this only provides approximate VCF, Figure 5 (left) shows that HSCIC achieves more counterfactually fair outcome distributions (more mass near zero) than an unconstrained classifier ($\gamma = 0$). Figure 5 (right) highlights once more that the HSCIC operator is in agreement with the VCF, again trading off accuracy.

## 5 DISCUSSION AND FUTURE WORK

We studied the problem of learning predictors $\hat{\mathbf{Y}}$ that are counterfactually invariant in changes of certain covariates. We put forth a formal definition of counterfactual invariance and described how it generalizes existing notions. Next, we provided a novel sufficient graphical criterion to characterize counterfactual invariance and reduce it to a conditional independence statement in the observational distribution. Our method does not require identifiability of the counterfactual distribution or exclude the possibility of unobserved confounders. Finally, we propose an efficiently estimable, model-agnostic regularization term to enforce this conditional independence (and thus counterfactual invariance) based on kernel mean embeddings of conditional measures, which works for mixed continuous/categorical, multi-dimensional variables. We demonstrate the efficacy of our method in regression and classification tasks involving controlled detailed simulation studies, high-dimensional images, and in a fairness application, where it outperforms existing baselines.

The main limitation of our work, shared by all studies in this domain, is the assumption that the causal graph is known. Another limitation is that our methodology is applicable only when our graphical criterion is satisfied, requiring a certain set of variables to be observed (albeit unobserved confounders are not generally excluded). From an ethics perspective, the increased robustness of counterfactually invariant (or societal benefits of counterfactually fair) predictors are certainly desirable. However, this presupposes that all the often untestable assumptions are valid. Overall, causal methodology should not be applied lightly, especially in high-stakes and consequential decisions. A critical analysis of the broader context or systemic factors may hold more promise for societal benefits, than a well-crafted algorithmic predictor.

In light of these limitations, an interesting and important direction for future work is to assess the sensitivity of our method to misspecifications of the causal graph or insufficient knowledge of the required blocking set. Finally, we believe our graphical criterion and KME-based regularization can also be useful for causal representation learning, where one aims to isolate causally relevant, autonomous factors underlying the data generating process of high-dimensional data.

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

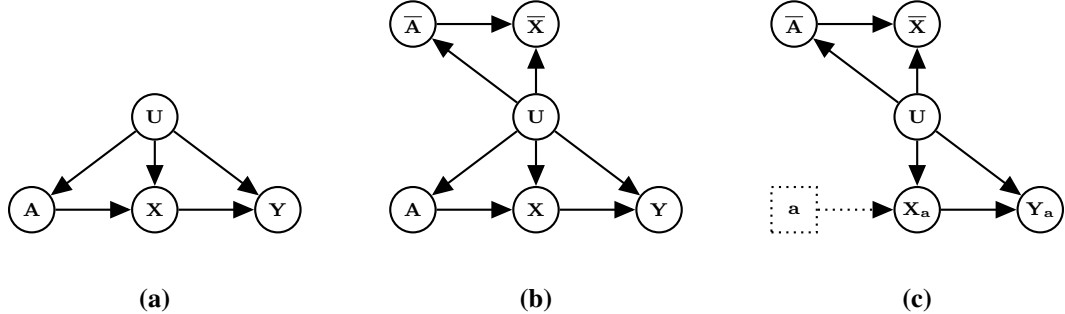

**(a)**                 **(b)**                 **(c)**

Figure 6: **(a) A causal graph** $\mathcal{G}$, which embeds information for the random variables of the model in the pre-interventional world. **(b) The corresponding graph** $\mathcal{G}'$ for the set $\mathbf{W} = \{\mathbf{A}, \mathbf{X}\}$. The variables $\overline{\mathbf{A}}$ and $\overline{\mathbf{X}}$ are *copies* of $\mathbf{A}$ and $\mathbf{X}$ respectively. **(c) The post-interventional graph** $\mathcal{G}'_{\mathbf{a}}$**.** By construction, any intervention of the form $\mathbf{A} \leftarrow \mathbf{a}$ does not affect the group $\overline{\mathbf{W}} = \{\overline{\mathbf{A}}, \overline{\mathbf{X}}\}$.

# A   MISSING PROOF OF THEOREM 3.2

In this section, we give proof of Theorem 3.2, which we restate for completeness.

## A.1   OVERVIEW OF THE PROOF TECHNIQUES

**Theorem 3.2.** *Let $\mathcal{G}$ be a causal diagram, and let $\mathbf{A}$, $\mathbf{W}$ be two (not necessarily disjoint) sets of nodes in $\mathcal{G}$. Let $\mathbf{Z}$ be a set of nodes that blocks all* non-causal[5] *paths from $\mathbf{A} \cup \mathbf{W}$ to $\mathbf{Y}$. Then, for any SCM compatible with $\mathcal{G}$, any predictor $\hat{\mathbf{Y}}$ that satisfies $\hat{\mathbf{Y}} \perp\!\!\!\perp \mathbf{A} \mid \mathbf{Z}$ is counterfactually invariant in $\mathbf{A}$ with respect to $\mathbf{W}$.*

Our proof technique generalizes the work of Shpitser & Pearl (2009). To understand the proof technique, note that conditional counterfactual distributions of the form $\mathbb{P}_{\mathbf{Y}^*_{\mathbf{a}} \mid \mathbf{W}}(\mathbf{y} \mid \mathbf{w})$ involve quantities *from two different worlds*. The variables $\mathbf{W}$ belong to the pre-interventional world, and the interventional variable $\mathbf{Y}^*_{\mathbf{a}}$ belongs to the world after performing the intervention intervention $\mathbf{A} \leftarrow \mathbf{a}$. Hence, we study the identification of conditional counterfactual distributions using a diagram that embeds the causal relationships between the pre- and the post-interventional world. After defining this diagram, we prove that some conditional measures in this new model provide an estimate for $\mathbb{P}_{\mathbf{Y}^*_{\mathbf{a}} \mid \mathbf{W}}(\mathbf{y} \mid \mathbf{w})$. We then combine this result with the properties of $\mathbf{Z}$ to prove the desired result.

## A.2   A GRAPHICAL CRITERION FOR CONDITIONAL INDEPENDENCE

In this section, we discuss a well-known criterion for conditional independence, which we will then use to prove Theorem 3.2. To this end, we use the notion of a blocked path, which we restate for clarity.

**Definition 3.1.** *Consider a path $\pi$ of causal graph $\mathcal{G}$. A set of nodes $\mathbf{Z}$ blocks $\pi$, if $\pi$ contains a triple of consecutive nodes connected in one of the following ways: $N_i \rightarrow Z \rightarrow N_j$, $N_i \leftarrow Z \rightarrow N_j$, or $N_i \rightarrow M \leftarrow N_j$, with $N_i, N_j \notin \mathbf{Z}$, $Z \in \mathbf{Z}$, and neither $M$ nor any descendent of $M$ is in $\mathbf{Z}$.*

Using Definition A.1, we define the concept of $d$-separation as follows.

**Definition A.1** ($d$-Separation)**.** *Consider a a causal graph $\mathcal{G}$. Two sets of nodes $\mathbf{X}$ and $\mathbf{Y}$ of $\mathcal{G}$ are said to be $d$-separated by a third set $\mathbf{Z}$ if every path from any node of $\mathbf{X}$ to any node of $\mathbf{Y}$ is blocked by $\mathbf{Z}$.*

Definition A.1 is a graphical criterion for conditional independence. In fact, the following well-known results holds (Pearl, 2000).

---

[5]A non-causal paths from $\mathbf{A} \cup \mathbf{W}$ to $\mathbf{Y}$ is a path connecting $\mathbf{A} \cup \mathbf{W}$ and $\mathbf{Y}$ in which at least one edge points against causal ordering. Shpitser et al. (2012)

**Lemma A.2** (*d*-Separation Criterion)**.** *Consider a a causal graph $\mathcal{G}$, and suppose that two sets of nodes $\mathbf{X}$ and $\mathbf{Y}$ of $\mathcal{G}$ are $d$-separated by $\mathbf{Z}$. Then, $\mathbf{X}$ is independent of $\mathbf{Y}$ given $\mathbf{Z}$ in any model induced by the graph $\mathcal{G}$.*

We use the notation $\mathbf{X} \perp\!\!\!\perp_{\mathcal{G}} \mathbf{Y} \mid \mathbf{Z}$ to indicate that $\mathbf{X}$ and $\mathbf{Y}$ are $d$-separated by $\mathbf{U}$ in $\mathcal{G}$.

## A.3    Identifiability of conditional counterfactual distributions

A natural way to study the relationships between the pre- and the post-interventional world is to use the counterfactual graph (Shpitser & Pearl, 2008). However, the construction of the counterfactual graph is rather intricate. For our purposes it is sufficient to consider a simpler construction. Consider an SGM with causal graph $\mathcal{G}$, and fix a set of observable random variables $\mathbf{W}$. We define the corresponding graph $\mathcal{G}'$ in the following three steps:

1. define $\mathcal{G}'$ to be the same graph as $\mathcal{G}$;
2. add a new node $\overline{W}$ to $\mathcal{G}'$, for each node $W$ of the set $\mathbf{W}$;
3. for each node $W$ of the set $\mathbf{W}$ and for each parent $W'$ of $W$, if $W' \in \mathbf{W}$ then add an edge $\overline{W} \rightarrow \overline{W}'$ to $\mathcal{G}'$; if $W' \notin \mathbf{W}$ add an edge $W' \rightarrow \overline{W}$ to $\mathcal{G}'$.

An illustration of this graph is presented in Figure 6. Note that any node $\overline{W}$ defined as above does not have any descendants in $\mathcal{G}'$. In the reminder of this section, we denote with $\overline{\mathbf{W}}$ the set of all nodes $\overline{W}$ in $\mathcal{G}'$ defined as above. We remark that this construction generalizes the work by Shpitser & Pearl (2009).

Fix a second set $\mathbf{A}$ of nodes of $\mathcal{G}$, and consider interventions of the form $\mathbf{A} \leftarrow \mathbf{a}$, as in the statement of Theorem 3.2. We prove that any conditional counterfactual distribution of the form $\mathbb{P}_{\mathbf{Y}_{\mathbf{a}}^*|\mathbf{W}}(\mathbf{y} \mid \mathbf{w})$ are identifiable in $\mathcal{G}$ if and only if the corresponding probability distributions $\mathbb{P}_{\mathbf{Y}|\overline{\mathbf{W}}}(\mathbf{y} \mid \overline{\mathbf{w}})$ are identifiable in $\mathcal{G}'_{\mathbf{a}}$. The following theorem generalizes Theorem 1 by Shpitser & Pearl (2009).

**Theorem A.3.** *Let $\mathcal{G}$ be a causal diagram, and consider two sets of nodes $\mathbf{A}$, $\mathbf{W}$ (not necessarily disjoint). Consider the corresponding graph $\mathcal{G}'$ as defined above. If the distribution $\mathbb{P}_{\mathbf{Y}_{\mathbf{a}}^*|\mathbf{W}}(\mathbf{y} \mid \mathbf{w})$ is identifiable in $\mathcal{G}$, then the distribution $\mathbb{P}_{\mathbf{Y}|\overline{\mathbf{W}}}(\mathbf{y} \mid \overline{\mathbf{w}})$ is identifiable in $\mathcal{G}'_{\mathbf{a}}$, for any model induced by $\mathcal{G}$. Furthermore, the estimand $\mathbb{P}_{\mathbf{Y}|\overline{\mathbf{W}}}(\mathbf{y} \mid \mathbf{w})$ in $\mathcal{G}'_{\mathbf{a}}$ is correct for $\mathbb{P}_{\mathbf{Y}_{\mathbf{a}}^*|\mathbf{W}}(\mathbf{y} \mid \mathbf{w})$.*

To prove Theorem A.3, we introduce additional concepts. We first introduce the notion of a $C$-forest and the notion of an edge (Shpitser & Pearl, 2006).

**Definition A.4** ($C$-Forest)**.** *Let $\mathcal{G}$ be a causal diagram, and consider a complete sub-graph $\mathcal{H}$ of $\mathcal{G}$. Denote with $\mathbf{R}$ the maximal root set of $\mathcal{H}$. We say that $\mathcal{H}$ is a $\mathbf{R}$-rooted C-forest if a subset of its bi-directed arcs forms a spanning tree over all vertices in $\mathcal{H}$, and all the observable nodes of $\mathcal{H}$ have at most one child.*

In our analysis we also use the following definition.

**Definition A.5** (Edge)**.** *Let $\mathcal{G}$ be a causal diagram, and fix a set of nodes $\mathbf{A}$. Consider two $\mathbf{R}$-rooted C-forests $\mathcal{H}$, $\mathcal{K}$ of $\mathcal{G}$ such that (i) $\mathcal{H}$ is a sub-graph of $\mathcal{K}$; (ii) $\mathcal{H}$ and $\mathcal{K}$ do not contain any variable in $\mathbf{A}$; (iii) the nodes of $\mathbf{R}$ are ancestors of $\mathbf{Y}$ in the graph $\mathcal{G}_{\mathbf{a}}$. Then, we say that $\mathcal{H}$ and $\mathcal{K}$ form an edge for $\mathbb{P}_{\mathbf{Y}_{\mathbf{a}}^*}(\mathbf{y})$ in $\mathcal{G}$.*

There is a connection between these concepts and the identifiability of counterfactual distributions, as shown in the following theorem.

**Theorem A.6** (Theorem 4 by Shpitser & Pearl (2006))**.** *Let $\mathcal{G}$ be a causal diagram, and fix a set of nodes $\mathbf{A}$. Suppose that there exist two sub-graphs of $\mathcal{G}$ that form an edge for $\mathbb{P}_{\mathbf{Y}_{\mathbf{a}}^*}(\mathbf{y})$. Then, $\mathbb{P}_{\mathbf{Y}_{\mathbf{a}}^*}(\mathbf{y})$ is not identifiable in $\mathcal{G}$.*

Using these concepts, we can prove Theorem A.3.

*Proof of Theorem A.3.* For simplicity, denote with $\mathbb{P}^*$ the induced measure on $\mathcal{G}'_{\mathbf{a}}$. We first prove that $\mathbb{P}_{\mathbf{Y}_{\mathbf{a}}^*|\mathbf{W}}(\mathbf{y} \mid \mathbf{w})$ is identifiable in $\mathcal{G}$ if and only if $\mathbb{P}^*_{\mathbf{Y}|\overline{\mathbf{W}}}(\mathbf{y} \mid \overline{\mathbf{w}})$ is identifiable in $\mathcal{G}'_{\mathbf{a}}$. We distinguish two cases, based on whether $\overline{\mathbf{W}}$ is $d$-separated from $\mathbf{Y}$ in $\mathcal{G}'_{\mathbf{a}}$ or not.

**Case 1: $\overline{\mathbf{W}}$ is $d$-separated from $\mathbf{Y}$ in $\mathcal{G}'_{\mathbf{a}}$.** By Lemma A.2 we have that $\mathbf{Y}$ is independent of $\overline{\mathbf{W}}$. Hence, $\mathbb{P}^*_{\mathbf{Y}|\overline{\mathbf{W}}}(\mathbf{y} \mid \mathbf{w})$ is identifiable if and only if

$$\mathbb{P}^*_{\mathbf{Y}|\overline{\mathbf{W}}}(\mathbf{y} \mid \mathbf{w}) = \mathbb{P}^*_{\mathbf{Y}}(\mathbf{y}) = \mathbb{P}_{\mathbf{Y}^*_{\mathbf{a}}}(\mathbf{y}) \tag{5}$$

is identifiable in $\mathcal{G}'_{\mathbf{a}}$. Suppose that $\mathbb{P}_{\mathbf{Y}^*_{\mathbf{a}}|\mathbf{W}}(\mathbf{y} \mid \mathbf{w})$ is not identifiable. Then, $\mathbb{P}_{\mathbf{Y}^*_{\mathbf{a}}}(\mathbf{y})$ is not identifiable, and $\mathbb{P}^*_{\mathbf{Y}|\overline{\mathbf{W}}}(\mathbf{y} \mid \mathbf{w})$ is also not identifiable by eq. (5).

**Case 2: $\overline{\mathbf{W}}$ is not $d$-separated from $\mathbf{Y}$ in $\mathcal{G}'_{\mathbf{a}}$.** Assume without loss of generality that any variable of $\mathbf{A} \cup \mathbf{W}$ is not a descendent of $\mathbf{Y}$ in $\mathcal{G}$ (otherwise it has no effect on $\mathbf{Y}$). Under this assumption, there exists a set of random variables $\overline{\mathbf{U}} \subseteq \overline{\mathbf{W}}$ such that there exists an edge for $\mathbb{P}_{(\mathbf{Y} \cup \overline{\mathbf{U}})^*_{\mathbf{a}}}(\mathbf{y}, \overline{\mathbf{u}}) = \mathbb{P}^*_{\mathbf{Y},\overline{\mathbf{U}}}(\mathbf{y}, \overline{\mathbf{u}})$ in $\mathcal{G}'$. It follows that $\mathbb{P}^*_{\mathbf{Y},\overline{\mathbf{U}}}(\mathbf{y}, \overline{\mathbf{u}})$ is not identifiable. Since $\overline{\mathbf{U}} \subseteq \overline{\mathbf{W}}$, it follows that the distribution $\mathbb{P}^*_{\mathbf{Y},\overline{\mathbf{W}}}(\mathbf{y}, \overline{\mathbf{w}})$ is also not identifiable.

We conclude by showing that the estimand expression is correct. To this end, note that since $\mathcal{G}'$ is just a causal diagram, the estimand of the post-interventional distribution $\mathbb{P}^*_{\mathbf{Y}|\overline{\mathbf{W}}}(\mathbf{y} \mid \mathbf{w})$ is correct for $\mathbb{P}_{\mathbf{Y}^*_{\mathbf{a},\mathbf{x}}|\mathbf{A},\mathbf{X}}(\mathbf{y} \mid \mathbf{a}', \mathbf{x})$. Since the set $\overline{\mathbf{W}}$ only contains variable copies of $\mathbf{A} \cup \mathbf{X}$, as claimed. $\qquad\square$

### A.4 Valid adjustments for conditional interventional distributions

Here we discuss a criterion for the identification of conditional distributions, which we will then use to prove Theorem 3.2. We follow Pearl (2000) to this end, and use the $d$-separation criterion to define valid adjustment sets for conditional counterfactual distributions.

Consider a model with causal graph $\mathcal{G}$, and fix a set of observed variables $\mathbf{A}$. We define an auxiliary graph $\mathcal{G}_{\mathbf{I}}$, by adding to $\mathcal{G}$ an additional node $\mathbf{I}$. This node is a parent of the nodes in the set $\mathbf{A}$, and it has no other neighbour. We modify the structural assignments of the nodes $\mathbf{A}$, so that for $\mathbf{I} = 0$ the values of $\mathbf{A}$ are determined as in $\mathcal{G}$, whereas for $\mathbf{I} = 1$ the values of $\mathbf{A}$ are set to $\mathbf{A} = \mathbf{a}$. The node $\mathbf{I}$ corresponds to a Bernoulli distribution, with $\mathbf{I} = 1$ indicating that the intervention took place. This construction is important, because the following lemma holds.

**Lemma A.7.** *Consider an SGM $\mathcal{G}$ and fix two disjoints groups of observed variables $\mathbf{X}$ and $\mathbf{Y}$. Denote with $\mathcal{G}_{\mathbf{I}}$ the auxiliary graph as defined above, with respect to an intervention $\mathbf{A} \leftarrow \mathbf{a}$. Let $\mathbf{Z}$ a set of nodes of $\mathcal{G}$ such that $\mathbf{Y} \perp\!\!\!\perp_{\mathcal{G}_{\mathbf{I}}} \mathbf{I} \mid \mathbf{A}, \mathbf{Z}$. Then, it holds*

$$\mathbb{P}^*_{\mathbf{Y}|\mathbf{Z}}(\mathbf{y} \mid \mathbf{z}) = \mathbb{P}_{\mathbf{Y}|\mathbf{A},\mathbf{Z}}(\mathbf{y} \mid \mathbf{a}, \mathbf{z}).$$

*Here, $\mathbb{P}^*$ the post-interventional distribution, after assigning $\mathbf{A} \leftarrow \mathbf{a}$.*

*Proof.* It holds

$$\begin{aligned}
\mathbb{P}^*_{\mathbf{Y}|\mathbf{Z}}(\mathbf{y} \mid \mathbf{z}) &= \mathbb{P}^*_{\mathbf{Y}|\mathbf{A},\mathbf{Z}}(\mathbf{y} \mid \mathbf{a}, \mathbf{z}) && \text{(by definition of } \mathbb{P}^* ) \\
&= \mathbb{P}_{\mathbf{Y}|\mathbf{I},\mathbf{A},\mathbf{Z}}(\mathbf{y} \mid \mathbf{i} = 1, \mathbf{a}, \mathbf{z}) && \text{(by the definition of } \mathbf{I}) \\
&= \mathbb{P}_{\mathbf{Y}|\mathbf{I},\mathbf{A},\mathbf{Z}}(\mathbf{y} \mid \mathbf{i} = 0, \mathbf{a}, \mathbf{z}) && \text{(by Lemma A.2)} \\
&= \mathbb{P}_{\mathbf{Y}|\mathbf{A},\mathbf{Z}}(\mathbf{y} \mid \mathbf{a}, \mathbf{z}), && \text{(by the definition of } \mathbf{I})
\end{aligned}$$

as claimed. $\qquad\square$

The following lemma characterizes all sets $\mathbf{Z}$ that fulfills the condition as in Lemma A.7.

**Lemma A.8.** *Consider an SGM $\mathcal{G}$, and denote with $\mathcal{G}_{\mathbf{I}}$ the auxiliary graph as defined above, with respect to an intervention $\mathbf{A} \leftarrow \mathbf{a}$. Let $\mathbf{Z}$ a set of nodes that blocks all non-causal paths from $\mathbf{A}$ to $\mathbf{Y}$. Then, it holds $\mathbf{Y} \perp\!\!\!\perp_{\mathcal{G}_{\mathbf{I}}} \mathbf{I} \mid \mathbf{A}, \mathbf{Z}$. In particular, it holds $\mathbb{P}^*_{\mathbf{Y}|\mathbf{Z}}(\mathbf{y} \mid \mathbf{z}) = \mathbb{P}_{\mathbf{Y}|\mathbf{A},\mathbf{Z}}(\mathbf{y} \mid \mathbf{a}, \mathbf{z})$.*

*Proof.* Denote with $\pi$ any path from $\mathbf{I}$ to $\mathbf{Y}$ in $\mathcal{G}_{\mathbf{I}}$. Since $\mathbf{I}$ is a parent of every node in $\mathbf{A}$, and since $\mathbf{I}$ has no parents, then any path $\pi$ can be decomposed into two paths $\pi_1, \pi_2$, where $\pi_1$ is a single edge from $\mathbf{I}$ to $\mathbf{A}$, and $\pi_2$ is a path from $\mathbf{A}$ to $\mathbf{Y}$. If $\pi$ is a causal path, then $\mathbf{A}$ acts as a separator for this path. If $\pi$ is not a causal path from $\mathbf{I}$ to $\mathbf{Y}$, then it must be undirected (since $\mathbf{I}$ has no parents). It follows that $\pi_2$ is an undirected path from $\mathbf{A}$ to $\mathbf{Y}$, which is $d$-separated by $\mathbf{Z}$. Hence, $\pi$ is $d$-separated. We conclude that $\mathbf{Y} \perp\!\!\!\perp_{\mathcal{G}_{\mathbf{I}}} \mathbf{I} \mid \mathbf{A}, \mathbf{Z}$. The second part of the claim follows by Lemma A.7. $\qquad\square$

## A.5 PROOF OF THEOREM 3.2

Theorem A.3 tells us that we can identify conditional counterfactual distributions in $\mathcal{G}$, by identifying distributions on $\mathcal{G}'$. We can combine this observation with the notion of a valid adjustment set to derive a closed formula for the identification of the distributions of interest.

*Proof of Theorem 3.2.* Let $\mathcal{G}'$ be the augmented graph obtained by adding nodes $\overline{\mathbf{W}}$ to $\mathcal{G}$, as described in Section A.3. Denote with $\mathbb{P}^*$ the induced measure on $\mathcal{G}'_{\mathbf{a}}$. Suppose that it holds

$$\mathbb{P}_{\mathbf{Y}^*_{\mathbf{a}}|\mathbf{W}}(\mathbf{y} \mid \mathbf{w}) = \int \mathbb{P}_{\mathbf{Y}|\mathbf{A},\mathbf{Z}}(\mathbf{y} \mid \mathbf{a}, \mathbf{z}) d\mathbb{P}^*_{\mathbf{Z}|\overline{\mathbf{W}}}(\mathbf{z} \mid \mathbf{w}) \tag{6}$$

for any intervention $\mathbf{A} \leftarrow \mathbf{a}$, and for any possible value $\mathbf{w}$ attained by $\mathbf{W}$. Then the claim follows. In fact, assuming that eq. (6) holds, we have that

$$\begin{aligned}
\mathbb{P}_{\mathbf{Y}^*_{\mathbf{a}}|\mathbf{W}}(\mathbf{y} \mid \mathbf{w}) &= \int \mathbb{P}_{\mathbf{Y}|\mathbf{A},\mathbf{Z}}(\mathbf{y} \mid \mathbf{a}, \mathbf{z}) d\mathbb{P}^*_{\mathbf{Z}|\overline{\mathbf{W}}}(\mathbf{z} \mid \mathbf{w}) && \text{(assuming eq. (6))} \\
&= \int \mathbb{P}_{\mathbf{Y}|\mathbf{A},\mathbf{Z}}(\mathbf{y} \mid \mathbf{a}', \mathbf{z}) d\mathbb{P}^*_{\mathbf{Z}|\overline{\mathbf{W}}}(\mathbf{z} \mid \mathbf{w}) && \text{(since } \mathbf{Y} \perp\!\!\!\perp \mathbf{A} \mid \mathbf{Z}) \\
&= \mathbb{P}_{\mathbf{Y}^*_{\mathbf{a}'}|\mathbf{W}}(\mathbf{y} \mid \mathbf{w}). && \text{(assuming eq. (6))}
\end{aligned}$$

Hence, the proof of Theorem 3.2 boils down to proving eq. (6). To this end, since $\mathbf{Z}$ breaks all non-causal paths from $\mathbf{W}$ to $\mathbf{Y}$, then by construction $\mathbf{Z}$ is a $d$-separator between $\overline{\mathbf{W}}$ and $\mathbf{Y}$ in the post-interventional graph $\mathcal{G}'_{\overline{\mathbf{a}}}$. Hence, it holds

$$\begin{aligned}
\mathbb{P}^*_{\mathbf{Y},\overline{\mathbf{W}}}(\mathbf{y} \mid \overline{\mathbf{w}}) &= \int \mathbb{P}^*_{\mathbf{Y}|\overline{\mathbf{W}},\mathbf{Z}}(\mathbf{y} \mid \mathbf{w}, \mathbf{z}) d\mathbb{P}^*_{\mathbf{Z}|\overline{\mathbf{W}}}(\mathbf{z} \mid \overline{\mathbf{w}}) && \text{(by conditioning)} \\
&= \int \mathbb{P}^*_{\mathbf{Y}|\mathbf{Z}}(\mathbf{y} \mid \mathbf{z}) d\mathbb{P}^*_{\overline{\mathbf{W}},\mathbf{Z}}(\mathbf{w}, \mathbf{z}) && \text{(since } \mathbf{Y} \perp\!\!\!\perp_{\mathcal{G}'_{\mathbf{a}}} \overline{\mathbf{W}} \mid \mathbf{Z}) \\
&= \int \mathbb{P}_{\mathbf{Y}|\mathbf{A},\overline{\mathbf{W}},\mathbf{Z}}(\mathbf{y} \mid \mathbf{a}, \overline{\mathbf{w}}, \mathbf{z}) d\mathbb{P}^*_{\mathbf{Z}|\overline{\mathbf{W}}}(\mathbf{z} \mid \overline{\mathbf{w}}). && \text{(by Lemma A.7)}
\end{aligned}$$

The claim follows by applying Theorem A.3 to the equation above, since it holds $\mathbb{P}^*_{\mathbf{Y}|\overline{\mathbf{W}}}(\mathbf{y} \mid \mathbf{w}) = \mathbb{P}_{\mathbf{Y}^*_{\mathbf{a}}|\mathbf{W}}(\mathbf{y} \mid \mathbf{w})$. $\qquad\square$

## B MISSING PROOF OF THEOREM 3.4

We prove that the HSCIC can be used to promote conditional independence, using a similar technique as Park & Muandet (2020). The following theorem holds.

**Theorem 3.4** (following Theorem 5.4 by Park & Muandet (2020)). *If the kernel $k$ of $\mathcal{H}_{\mathbf{X}} \otimes \mathcal{H}_{\mathbf{A}}$ is characteristic[6], $\mathrm{HSCIC}(\mathbf{Y}, \mathbf{A} \mid \mathbf{Z}) = 0$ almost surely if and only if $\mathbf{Y} \perp\!\!\!\perp \mathbf{A} \mid \mathbf{Z}$.*

*Proof.* By definition, we can write $\mathrm{HSCIC}(\mathbf{Y}, \mathbf{A} \mid \mathbf{Z}) = H_{\mathbf{Y},\mathbf{A}|\mathbf{Z}} \circ \mathbf{Z}$, where $H_{\mathbf{Y},\mathbf{A}|\mathbf{Z}}$ is a real-valued deterministic function. Hence, the HSCIC is a real-valued random variable, defined over the same domain $\Omega_{\mathbf{Z}}$ of the random variable $\mathbf{X}$.

We first prove that if $\mathrm{HSCIC}(\mathbf{Y}, \mathbf{A} \mid \mathbf{Z}) = 0$ almost surely, then it holds $\mathbf{Y} \perp\!\!\!\perp \mathbf{A} \mid \mathbf{Z}$. To this end, consider an event $\Omega' \subseteq \Omega_{\mathbf{X}}$ that occurs almost surely, and such that it holds $(H_{\mathbf{Y},\mathbf{A}|\mathbf{X}} \circ \mathbf{X})(\omega) = 0$ for all $\omega \in \Omega'$. Fix a sample $\omega \in \Omega'$, and consider the corresponding value $\mathbf{z}_\omega = \mathbf{Z}(\omega)$, in the support of $\mathbf{Z}$. It holds

$$\begin{aligned}
\int k(\mathbf{y} \otimes \mathbf{a}, \,\cdot\,) d\mathbb{P}_{\mathbf{Y},\mathbf{A}|\mathbf{Z}=\mathbf{z}_\omega} &= \mu_{\mathbf{Y},\mathbf{A}|\mathbf{Z}=\mathbf{z}_\omega} && \text{(by definition)} \\
&= \mu_{\mathbf{Y}|\mathbf{Z}=\mathbf{z}_\omega} \otimes \mu_{\mathbf{A}|\mathbf{Z}=\mathbf{z}_\omega} && \text{(since } \omega \in \Omega') \\
&= \int k_{\mathbf{Y}}(\mathbf{y}, \,\cdot\,) d\mathbb{P}_{\mathbf{Y}|\mathbf{Z}=\mathbf{z}_\omega} \otimes \int k_{\mathbf{A}}(\mathbf{a}, \,\cdot\,) d\mathbb{P}_{\mathbf{A}|\mathbf{Z}=\mathbf{z}_\omega} && \text{(by definition )} \\
&= \int k_{\mathbf{Y}}(\mathbf{y}, \,\cdot\,) \otimes k_{\mathbf{A}}(\mathbf{a}, \,\cdot\,) d\mathbb{P}_{\mathbf{Y}|\mathbf{Z}=\mathbf{z}_\omega}\mathbb{P}_{\mathbf{A}|\mathbf{Z}=\mathbf{z}_\omega}, && \text{(by Fubini's Theorem)}
\end{aligned}$$

---

[6]The tensor product kernel $k$ is characteristic if the mapping $\mathbb{P}_{\mathbf{Y},\mathbf{A}} \mapsto \mathbb{E}_{\mathbf{y},\mathbf{a}}[k(\,\cdot\,, \mathbf{y} \otimes \mathbf{a})]$ is injective.

with $k_{\mathbf{Y}}$ and $k_{\mathbf{A}}$ the kernels of $\mathcal{H}_{\mathbf{Y}}$ and $\mathcal{H}_{\mathbf{A}}$ respectively. Since the kernel $k$ of the tensor product space $\mathcal{H}_{\mathbf{Y}} \otimes \mathcal{H}_{\mathbf{A}}$ is characteristic, then the kernels $k_{\mathbf{Y}}$ and $k_{\mathbf{A}}$ are also characteristic. Hence, it holds $\mathbb{P}_{\mathbf{Y},\mathbf{A}|\mathbf{Z}=\mathbf{z}_\omega} = \mathbb{P}_{\mathbf{Y}|\mathbf{Z}=\mathbf{z}_\omega}\mathbb{P}_{\mathbf{A}|\mathbf{Z}=\mathbf{z}_\omega}$ for all $\omega \in \Omega'$. Since the event $\Omega'$ occurs almost surely, then $\mathbb{P}_{\mathbf{Y},\mathbf{A}|\mathbf{Z}=\mathbf{z}_\omega} = \mathbb{P}_{\mathbf{Y}|\mathbf{Z}=\mathbf{z}_\omega}\mathbb{P}_{\mathbf{A}|\mathbf{Z}=\mathbf{z}_\omega}$ almost surely, that is $\mathbf{Y} \perp\!\!\!\perp \mathbf{A} \mid \mathbf{Z}$.

Assume now that $\mathbf{Y} \perp\!\!\!\perp \mathbf{A} \mid \mathbf{Z}$. By definition there exists an event $\Omega'' \subseteq \Omega_{\mathbf{Z}}$ such that $\mathbb{P}_{\mathbf{Y},\mathbf{A}|\mathbf{Z}=\mathbf{z}_\omega} = \mathbb{P}_{\mathbf{Y}|\mathbf{Z}=\mathbf{z}_\omega}\mathbb{P}_{\mathbf{A}|\mathbf{Z}=\mathbf{z}_\omega}$ for all samples $\omega \in \Omega''$, with $\mathbf{z}_\omega = \mathbf{Z}(\omega)$. It holds

$$
\begin{aligned}
\mu_{\mathbf{Y},\mathbf{A}|\mathbf{Z}=\mathbf{z}_\omega} &= \int k(\mathbf{y} \otimes \mathbf{a}, \ \cdot \ )d\mathbb{P}_{\mathbf{Y},\mathbf{A}|\mathbf{Z}=\mathbf{z}_\omega} && \text{(by definition)} \\
&= \int k(\mathbf{y} \otimes \mathbf{a}, \ \cdot \ )d\mathbb{P}_{\mathbf{Y}|\mathbf{Z}=\mathbf{z}_\omega}\mathbb{P}_{\mathbf{A}|\mathbf{Z}=\mathbf{z}_\omega} && \text{(since } \omega \in \Omega') \\
&= \int k_{\mathbf{Y}}(\mathbf{y}, \ \cdot \ )k_{\mathbf{A}}(\mathbf{a}, \ \cdot \ )d\mathbb{P}_{\mathbf{Y}|\mathbf{Z}=\mathbf{z}_\omega}\mathbb{P}_{\mathbf{A}|\mathbf{Z}=\mathbf{z}_\omega} && \text{(by definition of } k) \\
&= \int k_{\mathbf{Y}}(\mathbf{y}, \ \cdot \ )d\mathbb{P}_{\mathbf{Y}|\mathbf{Z}=\mathbf{z}_\omega} \otimes \int k_{\mathbf{A}}(\mathbf{a}, \ \cdot \ )d\mathbb{P}_{\mathbf{A}|\mathbf{Z}=\mathbf{z}_\omega} && \text{(by Fubini's Theorem)} \\
&= \mu_{\mathbf{Y}|\mathbf{Z}=\mathbf{z}_\omega} \otimes \mu_{\mathbf{A}|\mathbf{Z}=\mathbf{z}_\omega}. && \text{(by definition)}
\end{aligned}
$$

The claim follows. $\qquad\square$

## C  CONDITIONAL KERNEL MEAN EMBEDDINGS AND THE HSCIC

The notion of conditional kernel mean embeddings has already been studied in the literature. We show that, under stronger assumptions, our definition is equivalent to the definition by Park & Muandet (2020).

### C.1  CONDITIONAL KERNEL MEAN EMBEDDINGS AND CONDITIONAL INDEPENDENCE

We show that, under stronger assumptions, the HSCIC can be defined using the Bochner conditional expected value. The Bochner conditional expected value is defined as follows.

**Definition C.1.** *Fix two random variables $\mathbf{Y}$, $\mathbf{Z}$ taking value in a Banach space $\mathcal{H}$, and denote with $(\Omega, \mathcal{F}, \mathbb{P})$ their joint probability space. Then, the Bochner conditional expectation of $\mathbf{Y}$ given $\mathbf{Z}$ is any $\mathcal{H}$-valued random variable $\mathbf{X}$ such that*

$$
\int_E \mathbf{Y}d\mathbb{P} = \int_E \mathbf{X}d\mathbb{P}
$$

*for all $E \in \sigma(\mathbf{Z}) \subseteq \mathcal{F}$, with $\sigma(\mathbf{Z})$ the $\sigma$-algebra generated by $\mathbf{Z}$. We denote with $\mathbb{E}\left[\mathbf{Y} \mid \mathbf{Z}\right]$ the Bochner expected value. Any random variable $\mathbf{X}$ as above is a version of $\mathbb{E}\left[\mathbf{Y} \mid \mathbf{Z}\right]$.*

The existence and almost sure uniqueness of the conditional expectation is shown in Dinculeanu (2000). Given a RKHS $\mathcal{H}$ with kernel $k$ over the support of $\mathbf{Y}$, Park & Muandet (2020) define the corresponding conditional kernel mean embedding as

$$
\mu_{\mathbf{Y}|\mathbf{Z}} \coloneqq \mathbb{E}\left[k(\cdot, \mathbf{y}) \mid \mathbf{Z}\right].
$$

Note that, according to this definition, $\mu_{\mathbf{Y}|\mathbf{Z}}$ is an $\mathcal{H}$-valued random variable, not a single point of $\mathcal{H}$. Park & Muandet (2020) use this notion to define the HSCIC as follows.

**Definition C.2** (The HSCIC according to Park & Muandet (2020))**.** *Consider (sets of) random variables $\mathbf{Y}$, $\mathbf{A}$, $\mathbf{Z}$, and consider two RKHS $\mathcal{H}_{\mathbf{Y}}$, $\mathcal{H}_{\mathbf{A}}$ over the support of $\mathbf{Y}$ and $\mathbf{A}$ respectively. The HSCIC between $\mathbf{Y}$ and $\mathbf{A}$ given $\mathbf{Z}$ is defined as the real-valued random variable*

$$
\omega \mapsto \left\|\mu_{\mathbf{Y},\mathbf{A}|\mathbf{Z}}(\omega) - \mu_{\mathbf{Y}|\mathbf{Z}}(\omega) \otimes \mu_{\mathbf{A}|\mathbf{Z}}(\omega)\right\|,
$$

*for all samples $\omega$ in the domain $\Omega_{\mathbf{Z}}$ of $\mathbf{Z}$. Here, $\|\cdot\|$ the metric induced by the inner product of the tensor product space $\mathcal{H}_{\mathbf{Y}} \otimes \mathcal{H}_{\mathbf{Z}}$.*

We show that, under more restrictive assumptions, Definition C.2 can be used to promote conditional independence. To this end, we use the notion of a regular version.

**Definition C.3** (Regular Version, following Definition 2.4 by Çinlar & ðCınlar (2011))**.** *Consider two random variables* $\mathbf{Y}$, $\mathbf{Z}$, *and consider the induced measurable spaces* $(\Omega_{\mathbf{Y}}, \mathcal{F}_{\mathbf{Y}})$ *and* $(\Omega_{\mathbf{Z}}, \mathcal{F}_{\mathbf{Z}})$. *A regular version $Q$ for* $\mathbb{P}_{\mathbf{Y}|\mathbf{Z}}$ *is a mapping* $Q\colon \Omega_{\mathbf{Z}} \times \mathcal{F}_{\mathbf{Y}} \to [0, +\infty]\colon (\omega, \mathbf{y}) \mapsto Q_\omega(\mathbf{y})$ *such that: (i) the map* $\omega \mapsto Q_\omega(\mathbf{x})$ *is* $\mathcal{F}_{\mathbf{A}}$-*measurable for all* $\mathbf{y}$*; (ii) the map* $\mathbf{y} \mapsto Q_\omega(\mathbf{y})$ *is a measure on* $(\Omega_{\mathbf{Y}}, \mathcal{F}_{\mathbf{Y}})$ *for all* $\omega$*; (iii) the function* $Q_\omega(\mathbf{y})$ *is a version for* $\mathbb{E}\left[\not\Vdash_{\{\mathbf{Y}=\mathbf{y}\}} \mid \mathbf{Z}\right]$.

The following theorem shows that the random variable as in Definition C.2 can be used to promote conditional independence.

**Theorem C.4** (Theorem 5.4 by Park & Muandet (2020))**.** *With the notation introduced above, suppose that the kernel $k$ of the tensor product space* $\mathcal{H}_{\mathbf{X}} \otimes \mathcal{H}_{\mathbf{A}}$ *is characteristic. Furthermore, suppose that* $\mathbb{P}_{\mathbf{Y},\mathbf{A}|\mathbf{X}}$ *admits a regular version. Then,* $\left\|\mu_{\mathbf{Y},\mathbf{A}|\mathbf{Z}}(\omega) - \mu_{\mathbf{Y}|\mathbf{Z}}(\omega) \otimes \mu_{\mathbf{A}|\mathbf{Z}}(\omega)\right\| = 0$ *almost surely if and only if* $\mathbf{Y} \perp\!\!\!\perp \mathbf{A} \mid \mathbf{Z}$.

Note that the assumption of the existence of a regular version is essential in Theorem C.4. In this work, HSCIC is not used for conditional independence testing but as a conditional independence measure.

## C.2 EQUIVALENCE WITH OUR APPROACH

The following theorem, shows that under the existence of a regular version, conditional kernel mean embeddings can be defined using the Bochner conditional expected value. To this end, we use the following theorem.

**Theorem C.5** (Following Proposition 2.5 by Çinlar & ðCınlar (2011))**.** *Following the notation introduced in Definition C.3, suppose that* $\mathbb{P}_{\mathbf{Y}|\mathbf{Z}}(\cdot \mid \mathbf{Z})$ *admits a regular version* $Q_\omega(\mathbf{y})$. *Consider a kernel $k$ over the support of* $\mathbf{Y}$. *Then, the mapping*

$$\omega \mapsto \int k(\cdot, \mathbf{y}) dQ_\omega(\mathbf{y})$$

*is a version of* $\mathbb{E}\left[k(\cdot, \mathbf{y}) \mid \mathbf{Z}\right]$.

As a consequence of Theorem C.5, we prove the following result.

**Lemma C.6.** *Fix two random variables* $\mathbf{Y}$, $\mathbf{Z}$. *Suppose that* $\mathbb{P}_{\mathbf{Y}|\mathbf{Z}}$ *admits a regular version. Denote with* $\Omega_{\mathbf{Z}}$ *the domain of* $\mathbf{Z}$. *Then, there exists a subset* $\Omega \subseteq \Omega_{\mathbf{Z}}$ *that occurs almost surely, such that* $\mu_{\mathbf{Y}|\mathbf{Z}}(\omega) = \mu_{\mathbf{Y}|\mathbf{Z}=\mathbf{Z}(\omega)}$ *for all* $\omega \in \Omega$. *Here,* $\mu_{\mathbf{Y}|\mathbf{Z}=\mathbf{Z}(\omega)}$ *is the embedding of conditional measures as in Section 2.*

*Proof.* Let $Q_\omega(\mathbf{y})$ be a regular version of $\mathbb{P}_{\mathbf{Y}|\mathbf{Z}}$. Without loss of generality we may assume that it holds $\mathbb{P}_{\mathbf{Y}|\mathbf{Z}}(\mathbf{y} \mid \{\mathbf{Z} = \mathbf{Z}(\omega)\}) = Q_\omega(\mathbf{y})$. By Theorem C.5 there exists an event $\Omega \subseteq \Omega_{\mathbf{Z}}$ that occurs almost surely such that

$$\mu_{\mathbf{Y}|\mathbf{Z}}(\omega) = \mathbb{E}[k(\mathbf{y}, \,\cdot\,) \mid \mathbf{Z}](\omega) = \int k(\mathbf{y}, \,\cdot\,) dQ_\omega(\mathbf{y}), \tag{7}$$

for all $\omega \in \Omega$. Then, for all $\omega \in \Omega$ it holds

$$\mu_{\mathbf{Y}|\mathbf{Z}}(\omega) = \int k(\mathbf{x}, \,\cdot\,) dQ_\omega(\mathbf{x}) \qquad\qquad \text{(it follows from eq. (7))}$$

$$= \int k(\mathbf{x}, \,\cdot\,) d\mathbb{P}_{\mathbf{X}|\mathbf{A}}(\mathbf{x} \mid \{\mathbf{A} = \mathbf{A}(\omega)\}) \qquad (Q_\omega(\mathbf{y}) = \mathbb{P}_{\mathbf{Y}|\mathbf{Z}}(\mathbf{y} \mid \{\mathbf{Z} = \mathbf{Z}(\omega)\}))$$

$$= \mu_{\mathbf{X}|\{\mathbf{A}=\mathbf{A}(\omega)\}}, \qquad\qquad \text{(by definition as in Section 2)}$$

as claimed. □

As a consequence of Lemma C.6, we can prove that the definition of the HSCIC by Park & Muandet (2020) is equivalent to ours. The following corollary holds.

**Corollary C.7.** *Consider (sets of) random variables* $\mathbf{Y}$, $\mathbf{A}$, $\mathbf{Z}$, *and consider two RKHS* $\mathcal{H}_{\mathbf{Y}}$, $\mathcal{H}_{\mathbf{A}}$ *over the support of* $\mathbf{Y}$ *and* $\mathbf{A}$ *respectively. Suppose that* $\mathbb{P}_{\mathbf{Y},\mathbf{A}|\mathbf{Z}}(\cdot \mid \mathbf{Z})$ *admits a regular version. Then, there exists a set* $\Omega \subseteq \Omega_{\mathbf{A}}$ *that occurs almost surely, such that*

$$\left\|\mu_{\mathbf{X},\mathbf{A}|\mathbf{Z}}(\omega) - \mu_{\mathbf{X}|\mathbf{Z}}(\omega) \otimes \mu_{\mathbf{A}|\mathbf{Z}}(\omega)\right\| = (H_{\mathbf{Y},\mathbf{A}|\mathbf{Z}} \circ \mathbf{Z})(\omega).$$

Here, $H_{\mathbf{Y},\mathbf{A}|\mathbf{Z}}$ is a real-valued deterministic function, defined as

$$H_{\mathbf{Y},\mathbf{A}|\mathbf{Z}}(\mathbf{z}) := \left\| \mu_{\mathbf{Y},\mathbf{A}|\mathbf{Z}=\mathbf{z}} - \mu_{\mathbf{Y}|\mathbf{Z}=\mathbf{z}} \otimes \mu_{\mathbf{A}|\mathbf{Z}=\mathbf{z}} \right\|,$$

and $\|\cdot\|$ is the metric induced by the inner product of the tensor product space $\mathcal{H}_{\mathbf{X}} \otimes \mathcal{H}_{\mathbf{A}}$.

We remark that the assumption of the existence of a regular version is essential in Corollary C.7.

# D   CONDITIONAL INDEPENDENCE AND THE CROSS-COVARIANCE OPERATOR

In this section, we show that under additional assumptions, our definition of conditional KMEs is equivalent to the definition based on the cross-covariance operator, under more restrictive assumptions.

The definition of KMEs based on the cross-covariance operator requires the use of the following well-known result.

**Lemma D.1.** *Fix two RKHS $\mathcal{H}_{\mathbf{X}}$ and $\mathcal{H}_{\mathbf{Z}}$, and let $\{\varphi_i\}_{i=1}^{\infty}$ and $\{\psi_j\}_{j=1}^{\infty}$ be orthonormal bases of $\mathcal{H}_{\mathbf{X}}$ and $\mathcal{H}_{\mathbf{Z}}$ respectively. Denote with $\mathsf{HS}(\mathcal{H}_{\mathbf{X}}, \mathcal{H}_{\mathbf{Z}})$ the set of Hilbert-Schmidt operators between $\mathcal{H}_{\mathbf{X}}$ and $\mathcal{H}_{\mathbf{Z}}$. There is an isometric isomorphism between the tensor product space $\mathcal{H}_{\mathbf{X}} \otimes \mathcal{H}_{\mathbf{Z}}$ and $\mathsf{HS}(\mathcal{H}_{\mathbf{X}}, \mathcal{H}_{\mathbf{Z}})$, given by the map*

$$T: \sum_{i=1}^{\infty} \sum_{j=1}^{\infty} c_{i,j} \varphi_i \otimes \psi_j \mapsto \sum_{i=1}^{\infty} \sum_{j=1}^{\infty} c_{i,j} \langle \,\cdot\,, \varphi_i \rangle_{\mathcal{H}_{\mathbf{X}}} \psi_j.$$

For a proof of this result see i.e., Park & Muandet (2020). This lemma allows us to define the cross-covariance operator between two random variables, using the operator $T$.

**Definition D.2** (Cross-Covariance Oprator). *Consider two random variables $\mathbf{X}$, $\mathbf{Z}$. Consider corresponding mean embeddings $\mu_{\mathbf{X},\mathbf{Z}}$, $\mu_{\mathbf{X}}$ and $\mu_{\mathbf{Z}}$, as defined in Section 3. The cross-covariance operator is defined as $\Sigma_{\mathbf{X},\mathbf{Z}} := T(\mu_{\mathbf{X},\mathbf{Z}} - \mu_{\mathbf{X}} \otimes \mu_{\mathbf{Z}})$. Here, $T$ is the isometric isomorphism as in Lemma D.1.*

It is well-known that the cross-covariance operator can be decomposed into the covariance of the marginals and the correlation. That is, there exists a unique bounded operator $\Lambda_{\mathbf{Y},\mathbf{Z}}$ such that

$$\Sigma_{\mathbf{Y},\mathbf{Z}} = \Sigma_{\mathbf{Y},\mathbf{Y}}^{1/2} \circ \Lambda_{\mathbf{Y},\mathbf{Z}} \circ \Sigma_{\mathbf{Z},\mathbf{Z}}^{1/2}$$

Using this notation, we define the *normalized conditional cross-covariance operator*. Given three random variables $\mathbf{Y}$, $\mathbf{A}$, $\mathbf{Z}$ and corresponding kernel mean embeddings, this operator is defined as

$$\Lambda_{\mathbf{Y},\mathbf{A}|\mathbf{Z}} := \Lambda_{\mathbf{Y},\mathbf{A}} - \Lambda_{\mathbf{Y},\mathbf{Z}} \circ \Lambda_{\mathbf{Z},\mathbf{A}}. \tag{8}$$

This operator was introduce by Fukumizu et al. (2007). The normalized conditional cross-covariance can be used to promote statistical independence, as shown in the following theorem.

**Theorem D.3** (Theorem 3 by Fukumizu et al. (2007)). *Following the notation introduced above, define the random variable $\ddot{\mathbf{A}} := (\mathbf{A}, \mathbf{Z})$. Let $\mathbb{P}_{\mathbf{Z}}$ be the distribution of the random variable $\mathbf{Z}$, and denote with $L^2(\mathbb{P}_{\mathbf{Z}})$ the space of the square integrable functions with probability $\mathbb{P}_{\mathbf{Z}}$. Suppose that the tensor product kernel $k_{\mathbf{Y}} \otimes k_{\mathbf{A}} \otimes k_{\mathbf{Z}}$ is characteristic. Furthermore, suppose that $\mathcal{H}_{\mathbf{Z}} + \mathbb{R}$ is dense in $L^2(\mathbb{P}_{\mathbf{Z}})$. Then, it holds*

$$\Lambda_{\mathbf{Y},\ddot{\mathbf{A}}|\mathbf{Z}} = 0 \quad \text{if and only if} \quad \mathbf{Y} \perp\!\!\!\perp \mathbf{A} \mid \mathbf{X}.$$

*Here, $\Lambda_{\mathbf{Y},\ddot{\mathbf{A}}|\mathbf{Z}}$ is an operator defined as in eq. (8).*

By Theorem D.3, the operator $\Lambda_{\mathbf{Y},\ddot{\mathbf{A}}|\mathbf{Z}}$ can also be used to promote conditional independence. However, our method is more straightforward since it requires less assumptions. In fact, Theorem D.3 requires to embed the variable $\mathbf{Z}$ in a RKHS. In contrast, our method only requires the embedding on the variables $\mathbf{Y}$ and $\mathbf{A}$.

# E EXPERIMENT SETTINGS

Additional information about the experiments is now provided. The interested reader may refer to the source code provided in the supplementary material. In all cases, the experiments were performed on an Apple M1 Pro. No external GPU sources were used for the experimental setup.

## E.1 DATASETS FOR MODEL PERFORMANCE WITH THE USE OF THE HSCIC

The first set of synthetic experiments involves three different dataset simulations. The data-generating mechanism corresponding to the results in Figure 3 is the following:

$$\mathbf{Z} \sim \mathcal{N}(0,1) \qquad \mathbf{A} = \mathbf{Z}^2 + \varepsilon_{\mathbf{A}}$$

$$\mathbf{X} = \exp\left\{-\frac{1}{5}\mathbf{A}\right\}\mathbf{A} + \sin(2\mathbf{Z}) + \frac{1}{5}\varepsilon_{\mathbf{X}}$$

$$\mathbf{Y} = \frac{1}{2}\exp\{-\mathbf{XZ}\} \cdot \sin(2\mathbf{XZ}) + 5\mathbf{A} + \frac{1}{5}\varepsilon_{\mathbf{Y}},$$

where $\varepsilon_{\mathbf{A}} \sim \mathcal{N}(0,1)$ and $\varepsilon_{\mathbf{Y}}, \varepsilon_{\mathbf{X}} \overset{i.i.d.}{\sim} \mathcal{N}(0,0.1)$.

In the first experiment, Figure 3 shows the results of feed-forward neural networks consisting of 8 hidden layers with 20 nodes each, connected with a rectified linear activation function (ReLU) and a linear final layer. Mini-batch size of 256 and the Adam optimizer with a learning rate of $10^{-3}$ for 100 epochs were used. We set the range of trade-off parameter $\gamma \in [0,1]$ for all the experiments except the comparison against baselines CF1 and CF2.

## E.2 DATASETS FOR COMPARISON WITH BASELINES

The simulation procedure for the Scenario 1 and Scenario 2 in Table 1 respectively are the following. Scenario 1:

$$\mathbf{Z} \sim \mathcal{N}(0,1) \qquad \mathbf{A} = \mathbf{Z}^2 + \varepsilon_{\mathbf{A}}$$

$$\mathbf{X} = \frac{1}{2}\mathbf{A} \cdot \varepsilon_{\mathbf{X}} + 2\mathbf{Z}$$

$$\mathbf{Y} = \frac{1}{2}\exp\{-\mathbf{XZ}\} \cdot \sin(2\mathbf{XZ}) + 5\mathbf{A} + \frac{1}{5}\varepsilon_{\mathbf{Y}},$$

where $\varepsilon_{\mathbf{A}}, \varepsilon_{\mathbf{X}} \overset{i.i.d.}{\sim} \mathcal{N}(0,1)$ and $\varepsilon_{\mathbf{Y}} \overset{i.i.d.}{\sim} \mathcal{N}(0,0.1)$. Scenario 2:

$$\mathbf{Z} \sim \mathcal{N}(0,1) \qquad \mathbf{A} = \mathbf{Z}^2 + \varepsilon_{\mathbf{A}}$$

$$\mathbf{X} = \frac{1}{5}\mathbf{A} \cdot \varepsilon_{\mathbf{X}} + 2\exp\left\{-\frac{1}{2}\mathbf{Z}^2\right\}$$

$$\mathbf{Y} = \exp\{-\mathbf{Z}^2\} + \mathbf{AX} + \frac{1}{5}\varepsilon_{\mathbf{Y}},$$

where $\varepsilon_{\mathbf{A}}, \varepsilon_{\mathbf{X}} \overset{i.i.d.}{\sim} \mathcal{N}(0,1)$ and $\varepsilon_{\mathbf{Y}} \overset{i.i.d.}{\sim} \mathcal{N}(0,0.1)$.

Analysing the results in Table 1, the same hyperparameters as in the previous setting. This also holds for the baseline methods CF1 and CF2. In this table, both Scenario 1 and Scenario 2 were considered. The results shown in Figure 3 and Table 1 are the average and standard deviation resulting from respectively 10 and 4 random seeds runs.

E.3 DATASETS FOR MULTI-DIMENSIONAL VARIABLES EXPERIMENTS

The data-generating mechanisms for the multi-dimensional settings of Tables 2 are now shown. Analysing the results in Table 2 (top), given dimA $= D_1 \geq 2$, the datasets were generated from:

$$\mathbf{Z} \sim \mathcal{N}(0,1) \qquad \mathbf{A}_i = \mathbf{Z}^2 + \varepsilon_{\mathbf{A}}^i \quad \text{for } i \in \{1, D_1\}$$

$$\mathbf{X} = \exp\left\{-\frac{1}{2}\mathbf{A}_1\right\} + \sum_{i=1}^{D_1} \mathbf{A}_i \cdot \sin(\mathbf{Z}) + 0.1 \cdot \varepsilon_{\mathbf{X}}$$

$$\mathbf{Y} = \exp\left\{-\frac{1}{2}\mathbf{A}_2\right\} \cdot \sum_{i=1}^{D_1} \mathbf{A}_i + \mathbf{X}\mathbf{Z} + 0.1 \cdot \varepsilon_{\mathbf{Y}},$$

where $\varepsilon_{\mathbf{X}}, \varepsilon_{\mathbf{Y}} \overset{i.i.d}{\sim} \mathcal{N}(0, 0.1)$ and $\varepsilon_{\mathbf{A}}^1, ..., \varepsilon_{\mathbf{A}}^{D_1} \overset{i.i.d}{\sim} \mathcal{N}(0,1)$. In this setting, the mini-batch size chosen is 64 and the same hyperparameters are used as in the previous setting. The neural network architecture is trained for 70 epochs.

For the results in Table 2 (bottom) the following data-generating process is used:

$$\mathbf{Z}_1, \mathbf{Z}_2, ..., \mathbf{Z}_{D_2} \overset{i.i.d.}{\sim} \mathcal{N}(0,1) \qquad \mathbf{A} = \sum_{i=1}^{D_2} \mathbf{Z}_i^2 + \varepsilon_{\mathbf{A}}$$

$$\mathbf{X} = \exp\left\{-\frac{1}{2}\mathbf{A}\right\} + \sin\left(\sum_{i=1}^{D_2} \mathbf{Z}_i\right) \cdot \mathbf{A} + 0.1 \cdot \varepsilon_{\mathbf{X}}$$

$$\mathbf{Y} = \exp\left\{-\frac{1}{2}\mathbf{A}\right\} \cdot \mathbf{A} + \sum_{i=1}^{D_2} \mathbf{Z}_i + \mathbf{A} + \mathbf{X}\mathbf{Z}_1 + 0.1 \cdot \varepsilon_{\mathbf{Y}},$$

with dimZ $= D_2 \geq 2$ and $\varepsilon_{\mathbf{A}} \sim \mathcal{N}(0,1)$, $\varepsilon_{\mathbf{X}}, \varepsilon_{\mathbf{Y}} \overset{i.i.d.}{\sim} \mathcal{N}(0, 0.1)$. Here, we used mini-batch size of 32, a learning rate of $10^{-4}$ and a number of epochs of 500.

The results in Tables 2 are the average obtained from three random seeds runs on the same data-split.

We tested the method on a further setting, consisting of bi-dimensional $\mathbf{Z}$ and $\mathbf{A}$ (dimA $= 2$, dimZ $= 2$). Specifically, we have $\mathbf{Z} = \{\mathbf{Z}_1, \mathbf{Z}_2\}$ and $\mathbf{A} = \{\mathbf{A}_1, \mathbf{A}_2\}$. The data-generating mechanism is the following:

$$\mathbf{Z}_1 \sim \mathcal{N}(0,1) \quad \mathbf{Z}_2 \sim \mathcal{N}(3, 0.1) \quad \mathbf{A}_1 = \mathbf{Z}_1^2 + \varepsilon_{\mathbf{A}_1} \quad \mathbf{A}_2 = \exp\left\{-0.1 \cdot (\mathbf{Z}_1 + \mathbf{Z}_2)\right\} + \varepsilon_{\mathbf{A}_2}$$

$$\mathbf{X} = \exp\left\{-\frac{1}{2}\mathbf{A}\right\} \cdot \sin(2 \cdot \mathbf{A}_1) + (\mathbf{Z}_1 + \mathbf{Z}_2) \cdot (\mathbf{A}_1 + \mathbf{A}_2) + 0.1 \cdot \varepsilon_{\mathbf{X}}$$

$$\mathbf{Y} = \exp\left\{-\frac{1}{2}\mathbf{A}_1^2\right\} \cdot \sin\left(2 \cdot \mathbf{A}_2^2\right) + \mathbf{X} \cdot (\mathbf{Z}_1 + \mathbf{Z}_2) + 5 \cdot \mathbf{A}_1 \cdot \mathbf{A}_2 + 0.1 \cdot \varepsilon_{\mathbf{Y}},$$

where $\varepsilon_{\mathbf{Y}}, \varepsilon_{\mathbf{X}}, \varepsilon_{\mathbf{A}_1}, \varepsilon_{\mathbf{A}_2} \overset{i.i.d}{\sim} \mathcal{N}(0, 0.1)$. In Table 3, the trade-off between accuracy and counter-factually invariant predictions is once again shown, implying that the proposed method can also be applied in settings where both $\mathbf{Z}$ and $\mathbf{A}$ are not unidimensional. In Table 3 the average and standard deviation of the results from four runs with random seed and re-sampled data are presented.

Table 3: **Results of MSE, HSCIC, VCF (all times $10^5$ for readability) on synthetic data** with bi-dimensional **A** and **Z**. Here $\dim Z = 2, \dim A = 2, \dim X = 1, \dim Y = 1$.

|  | MSE | HSCIC | VCF |
|---|---|---|---|
| $\gamma = 0$ | $0.315 \pm 0.055$ | $13854.1 \pm 8.47$ | $24.6 \pm 1.17$ |
| $\gamma = \frac{1}{2}$ | $0.336 \pm 0.0525$ | $13854.0 \pm 8.46$ | $22.9 \pm 0.69$ |
| $\gamma = 1$ | $0.393 \pm 0.162$ | $13853.9 \pm 8.46$ | $20.4 \pm 2.02$ |

### E.4 HIGH-DIMENSIONAL IMAGE DATASET

The simulation procedure for the results shown in Section 4.2 is the following.

$$\texttt{shape} \sim \mathbb{P}(\texttt{shape})$$
$$\texttt{y-pos} \sim \mathbb{P}(\texttt{y-pos})$$
$$\texttt{color} \sim \mathbb{P}(\texttt{color})$$
$$\texttt{orientation} \sim \mathbb{P}(\texttt{orientation})$$
$$\texttt{x-pos} = \text{round}(x), \quad \text{where } x \sim \mathcal{N}(\texttt{shape} + \texttt{y-pos}, 1)$$
$$\texttt{scale} = \text{round}\Big( \Big( \frac{\texttt{x-pos}}{24} + \frac{\texttt{y-pos}}{24} \Big) \cdot \texttt{shape} + \epsilon_S \Big)$$
$$\mathbf{Y} = e^{\texttt{shape}} \cdot \texttt{x-pos} + \texttt{scale}^2 \cdot \sin(\texttt{y-pos}) + \epsilon_Y,$$

where $\epsilon_S \sim \mathcal{N}(0, 1)$ and $\epsilon_Y \sim \mathcal{N}(0, 0.01)$. The data has been generated via a matching procedure on the original dSprites dataset.

In Table 4, the hyperparameters of the layers of the convolutional neural network are presented. Each of the convolutional groups also has a ReLU activation function and a dropout layer. Two MLP architectures have been used. The former takes as input the observed tabular features. It is composed by two hidden layers of 16 and 8 nodes respectively, connected with ReLU activation functions and dropout layers. The latter takes as input the concatenated outcomes of the CNN and the other MLP. It consists of three hidden layers of 8, 8 and 16 nodes, respectively. Figure 4 presents the averaged results of four random seeds runs with new sampled data.

### E.5 FAIRNESS WITH CONTINUOUS PROTECTED ATTRIBUTES

The pre-processing of the UCI Adult dataset was based upon the work of (Chiappa & Pacchiano, 2021). Referring to the causal graph in Figure 7, a variational autoencoder (Kingma & Welling, 2014) was trained for each of the unobserved variables $\mathbf{H_m}$, $\mathbf{H_l}$ and $\mathbf{H_r}$. The prior distribution of these latent variables is assumed to be standard Gaussian. The posterior distributions $\mathbb{P}(\mathbf{H_m}|V)$, $\mathbb{P}(\mathbf{H_r}|V)$, $\mathbb{P}(\mathbf{H_l}|V)$ are modelled as 10-dimensional Gaussian distributions, whose means and variances are the outputs of the encoder.

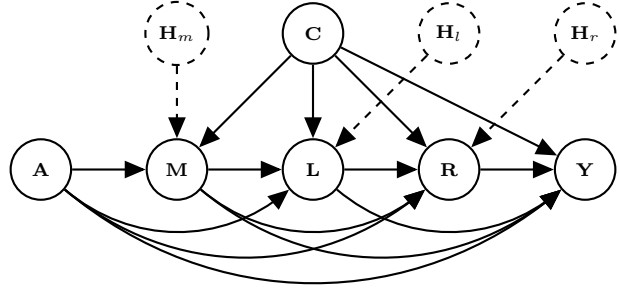

Figure 7: **Assumed causal graph for the Adult dataset**, as in Chiappa & Pacchiano (2021). The variables $\mathbf{H}_m$, $\mathbf{H}_l$, $\mathbf{H}_r$ are unobserved, and jointly trained with the predictor $\hat{\mathbf{Y}}$.

Table 4: **Architecture of the convolutional neural network** used for the image dataset, as described in Appendix E.4.

| layer | # filters | kernel size | stride size | padding size |
|---|---|---|---|---|
| convolution | 16 | 5 | 2 | 2 |
| max pooling | 1 | 3 | 2 | 0 |
| convolution | 64 | 5 | 1 | 2 |
| max pooling | 1 | 1 | 2 | 0 |
| convolution | 64 | 5 | 1 | 2 |
| max pooling | 1 | 2 | 1 | 0 |
| convolution | 16 | 5 | 1 | 3 |
| max pooling | 1 | 2 | 2 | 0 |

The encoder architecture consists of a hidden layer of 20 hidden nodes with hyperbolic tangent activation functions, followed by a linear layer. The decoders have two linear layers with hyperbolic tangent activation function. The training loss of the variational autoencoder consists of a reconstruction term (Mean-Squared Error for continuous variables and Cross-Entropy Loss for binary ones) and the Kullback–Leibler divergence between the posterior and the prior distribution of the latent variables. For training, we used the Adam optimizer with learning rate of $10^{-2}$, 30 epochs, mini-batch size 128.

The predictor $\hat{\mathbf{Y}}$ is the output of a feed-forward neural network consisting of a hidden layer with hyperbolic tangent activation function and a linear final layer. In the training we used the Adam optimizer with learning rate $10^{-3}$, mini-batch size 128, and trained for 100 epochs. The choice of the network architecture is based on the work of (Chiappa & Pacchiano, 2021).

The estimation of counterfactual outcomes is based on a Monte-Carlo approach. Given a data point, 500 values of the unobserved variables are sampled from the estimated posterior distribution. Given an interventional value for $A$, a counterfactual outcome is estimated for each of the sampled unobserved values. The final counterfactual outcome is estimated as the average of these counterfactual predictions. In this experiment setting, we have $k = 100$ and $d = 1000$.

In the causal graph presented in Figure 7, $\mathbf{A}$ includes the variables age and gender, $\mathbf{C}$ includes nationality and race, $\mathbf{M}$ marital status, $\mathbf{L}$ level of education, $\mathbf{R}$ the set of working class, occupation, and hours per week and $\mathbf{Y}$ the income class. Compared to (Chiappa & Pacchiano, 2021), we include the race variable in the dataset as part of the baseline features $\mathbf{C}$. The loss function is the same as Equation 2 but Binary Cross-Entropy loss is used instead of Mean-Squared Error loss:

$$\mathcal{L}_{\text{TOTAL}}(\hat{\mathbf{Y}}) = \mathcal{L}_{\mathbf{BCE}}(\hat{\mathbf{Y}}) + \gamma \cdot \text{HSIC}\left(\hat{\mathbf{Y}}, \{\text{Age}, \text{Gender}\}\Big|\mathbf{Z}\right), \tag{9}$$

where the set $\mathbf{Z}$ blocks all the non-causal paths from $\mathbf{W} \cup \mathbf{A}$. In this example we have $\mathbf{W} = \{\mathbf{C} \cup \mathbf{M} \cup \mathbf{L} \cup \mathbf{R}\}$. The results in Figure 5 (center, right) refer to one run with conditioning set $\mathbf{Z} = \{\text{Race}, \text{Nationality}\}$. The results in Table 5 (right) are the average and standard deviation of four random seeds.

### E.6 BASELINE EXPERIMENTS

We provide an experimental comparison against the method by Veitch et al. (2021). To this end, we consider the following artificial causal structure (see Figure 1(b)):

$$\mathbf{Z} \sim \mathcal{N}(0, 1) \qquad \mathbf{A} = \mathbf{Z}^2 + \varepsilon_{\mathbf{A}}$$

$$\mathbf{X} = \exp\left\{-\frac{1}{2}\mathbf{A}\right\} \sin(\mathbf{A}) + \frac{1}{10}\varepsilon_{\mathbf{X}}$$

$$\mathbf{Y} = \frac{1}{2}\exp\left\{-\mathbf{X}\mathbf{Z}\right\} \cdot \sin(2\mathbf{X}\mathbf{Z}) + 5\mathbf{A} + \frac{1}{10}\varepsilon_{\mathbf{Y}},$$

Table 5: Results of the **MSE**, HSCIC, **VCF** of our method and the baseline Veitch et al. (2021) applied to the causal and anti-causal structure in Figure 1(b-c). Although the graphical assumptions are not satisfied, our method shows an overall decrease of HSCIC, **VCF** in both of the graphical structures, outperforming Veitch et al. (2021) in terms of accuracy and counterfactual invariance.

| | **MSE** $\times 10^3$ | **HSCIC** $\times 10^3$ | **VCF** $\times 10^2$ |
|---|---|---|---|
| $\gamma = 0$ | $1.20 \pm 0.03$ | $22.88 \pm 1.42$ | $89 \pm 0.50$ |
| $\gamma = \frac{1}{2}$ | $3.72 \pm 0.17$ | $21.14 \pm 1.50$ | $89 \pm 0.30$ |
| $\gamma = 1$ | $10.45 \pm 0.35$ | $18.62 \pm 1.43$ | $87 \pm 0.93$ |

| | Our Method | | | Veitch et al. (2021) | | |
|---|---|---|---|---|---|---|
| | **MSE** $\times 10^2$ | **HSCIC** $\times 10^3$ | **VCF** $\times 10^2$ | **MSE** $\times 10^2$ | **HSCIC** $\times 10^3$ | **VCF** $\times 10^2$ |
| $\gamma = \frac{1}{2}$ | $69.78 \pm 1.10$ | $20.20 \pm 0.53$ | $50.63 \pm 0.25$ | $70.23 \pm 1.14$ | $22.16 \pm 0.57$ | $50.69 \pm 0.26$ |
| $\gamma = 1$ | $69.99 \pm 1.10$ | $18.81 \pm 0.53$ | $50.78 \pm 0.26$ | $70.60 \pm 1.24$ | $19.59 \pm 0.58$ | $50.85 \pm 0.28$ |

where $\varepsilon_{\mathbf{X}}, \varepsilon_{\mathbf{A}} \overset{i.i.d}{\sim} \mathcal{N}(0,1)$ and $\varepsilon_{\mathbf{Y}} \overset{i.i.d}{\sim} \mathcal{N}(0, 0.1)$. The data-generating mechanism of the anti-causal structure is the following (see Figure 1(c)):

$$\mathbf{Z} \sim \mathcal{N}(0,1)$$

$$\mathbf{Y} = \exp\left\{\frac{1}{5}\mathbf{Z}\right\} + \frac{3}{10}\varepsilon_{\mathbf{Y}} \qquad\qquad \mathbf{A} = \mathbf{Z}^2 + \frac{3}{10}\varepsilon_{\mathbf{A}}$$

$$\mathbf{X} = \exp\left\{-\frac{1}{2}\mathbf{A}^2\right\} + \frac{1}{5}\mathbf{Y} + \frac{1}{10}\varepsilon_{\mathbf{X}}$$

where $\varepsilon_{\mathbf{Y}}, \varepsilon_{\mathbf{A}} \overset{i.i.d}{\sim} \mathcal{N}(0, 0.1)$ and $\varepsilon_{\mathbf{X}} \overset{i.i.d}{\sim} \mathcal{N}(0,1)$. We compare our method with different choices for the trade-off parameter $\gamma$, against the method by Veitch et al. (2021). In the causal settings presented in Figure 1(b-c), an unobserved confounder $\mathbf{Z}$ between $\mathbf{A}$ and $\mathbf{Y}$ is included. In the graphical structure Figure 1(b), our method presents as regularization term in the model training $\mathrm{HSIC}(\hat{\mathbf{Y}}, \mathbf{A})$, as the independence $\hat{\mathbf{Y}} \perp\!\!\!\perp \mathbf{A}$ is enforced. Here, HSIC is the Hilbert-Schmidt Independence Criterion, which is commonly used to promote independence (see, i.e., Gretton et al. (2005); Fukumizu et al. (2007)). In the selected graphical sstructure, this is the same independence criterion enforced by Veitch et al. (2021), leading the two methods to converge. In the anti-causal graphical setting presented in Figure 1(c) proposed by Veitch et al. (2021), the regularization term used in our method is sill $\mathrm{HSIC}(\hat{\mathbf{Y}}, \mathbf{A})$, while in the method of Veitch et al. (2021) is $\mathrm{HSCIC}(\hat{\mathbf{Y}}, \mathbf{A} \mid \mathbf{Y})$. In Table 5, the results of accuracy, $\mathrm{HSCIC}(\hat{\mathbf{Y}}, \mathbf{A} \mid \mathbf{Z})$ and **VCF** are presented.

In the experiments, the predictor $\hat{\mathbf{Y}}$ is a feed-forward neural network consisting of 8 hidden layers with 20 nodes each, connected with a rectified linear activation function (ReLU) and a linear final layer. Mini-batch size of 256 and the Adam optimizer with a learning rate of $10^{-4}$ for 500 epochs were used.

### E.7 COMPARISON HEURISTIC METHODS EXPERIMENTS

We provide an experimental comparison of the proposed method with some heuristic methods, specifically data-augmentation based methods. We consider the same data-generating procedure and causal structure as presented in E. The heuristic methods considered are *data augmentation* and *causal-based data augmentation*. In the former, data augmentation is performed by generating $N = 50$ samples for every data-point by sampling new values of $\mathbf{A}$ as $a_1, ..., a_N \overset{i.i.d}{\sim} \mathbb{P}_{\mathbf{A}}$ and leaving $\mathbf{Z}, \mathbf{X}, \mathbf{Y}$ **unchanged**. Differently, in the latter *causal-based data augmentation* method, we also take into account the causal structure given by the known DAG. Indeed, when manipulating the variable $\mathbf{A}$, its descendants (in this example $\mathbf{X}$) will also change. In this experiment, a predictor for $\mathbf{X}$ as $\hat{\mathbf{X}} = f_\theta(\mathbf{A}, \mathbf{Z})$ is trained on $80\%$ of the original dataset. In the data augmentation mechanism, for every data-point $\{a, x, z, y\}$, $N = 50$ samples are generated by sampling new values of $\mathbf{A}$ as

Table 6: **Results of MSE and VCF (all times $10^2$ for readability) on synthetic data** of our method with trade-off parameters $\gamma = \frac{1}{2}$ and $\gamma = 1$ with the heuristic methods data augmentation and causal-based data augmentation.

|  | VCF | MSE |
| --- | --- | --- |
| data augmentation | $3.12 \pm 0.16$ | $0.003 \pm 0.001$ |
| causal-based data augmentation | $3.04 \pm 0.16$ | $0.013 \pm 0.012$ |
| $\gamma = \frac{1}{2}$ | $2.80 \pm 0.13$ | $0.044 \pm 0.022$ |
| $\gamma = 1$ | $2.34 \pm 0.19$ | $0.19 \pm 0.072$ |

$a_1, ..., a_N \overset{i.i.d}{\sim} \mathbb{P}_{\mathbf{A}}$, estimating the values of $\mathbf{X}$ as $x_1 = f_\theta(a_1, z), ..., x_N = f_\theta(a_N, z)$, while leaving the values of $\mathbf{Z}$ and $\mathbf{Y}$ unchanged. Heuristic methods such as data-augmentation methods do not theoretically guarantee to provide counterfactually invariant predictors. The results of an empirical comparison are shown in Table 6. It can be shown that these theoretical insights are supported by experimental results, as the **VCF** metric measure counterfactual invariance is relevantly lower in both of the two settings of the proposed methods ($\gamma = \frac{1}{2}$ and $\gamma = 1$).

In these experiments, a dataset of $n = 1000$ is used, along with $k = 500$ and $d = 500$. The architecture used for predicting $\mathbf{X}$ and $\mathbf{Y}$ are feed-forward neural networks consisting of 8 hidden layers with 20 nodes each, connected with a rectified linear activation function (ReLU) and linear final layer. Mini-batch size of 256 and the Adam optimizer with a learning rate of $10^{-3}$ for 100 epochs were used.

