# OpenReview forum: "Learning Counterfactually Invariant Predictors"
_ICLR.cc/2023/Conference — Submitted to ICLR 2023_

### Official Review · Reviewer_yLyi · 2022-10-23

**Confidence:** 2
**Correctness:** 3
**Technical Novelty And Significance:** 2
**Empirical Novelty And Significance:** 3
**Recommendation:** 6

**Clarity, Quality, Novelty And Reproducibility:**

The paper is clearly written. I am not quite familiar with related studies about invariant representation learning. For me, the proposed counterfactual invariance is novel and makes sense. And the authors make enough efforts to incorporate it into the learning process in practice. The most interesting thing is that their method can apply even if the counterfactual is not identifiable, which is surprising.

Questions:
My main question is how to use counterfactual invariance in the learning process. As far as I understand, the optimization object is Eq. 2, where the authors want to obtain a model to obtain $\hat{Y}$ which is conditional independent of $A$ given $\mathbf{Z}$. And the realization of HSCIC in Eq. 2 is Eq. 3.  But Eq. 3 seems to be estimating the conditional independence between $Y$ and $A$ given $\mathbf{Z}$, where $Y$ is the true label in the sample. Shouldn't they have been conditional independent because you assume there is a pre-known causal graph and in the causal graph they are conditional independent? Why isn't it $\hat{H}_{\hat{Y},A\mid \mathbf{Z}}$, where $\hat{Y}$ denotes the predicted label of the training model?

One detail:
Interventional distribution differs in general from the conditional distribution due to possible unobserved confounding effects. It is right. But "possible unobserved confounding effects" is just one possible reason. In fact, they can differ even if there are only observed confounding effects. Hence I suggest another statement.

**Strength And Weaknesses:**

Strength:
1. It is a very impressive result that even if the counterfactual is not identifiable, the counterfactual invariance can still be achieved.
2. The paper is clearly written.
3. The proposed counterfactual invariance is very novel

Weaknesses:
Currently I could not give any specific weaknesses of this paper, but only some questions. See the questions in the next part.

**Summary Of The Paper:**

In this paper, the authors propose counterfactual invariance, which is more general than some previous invariance objects. They show that the counterfactual invariance is equivalent to a conditional independence relation. Hence, by restricting conditional independence, the predicted target is counterfactual invariance. In the implementation, they add the restriction on conditional independence into the loss function, which can be achieved by HSICC.

**Summary Of The Review:**

#-----------------After rebuttal----------------------#

I want to thank the authors for their rebuttal. For me, I think the proposed method is novel, and the paper indeed provides a good perspective for "invariant predictors". After reading other reviewers' comments, I realize that there are some parts unclear or unsuitable right now. More importantly, I think the anonymous comments make sense. Although the authors give responses, it is not very clear to me. I think this part is quite important and needs some more discussion, which possibly needs another round of review if there are no new discussions from the authors or other reviewers. Hence I adjust my score right now.


#--------------------------#


Overall, the paper is of high quality. The contribution is novel and the writing is clear. I like this paper and I am happy to see it accepted.

---

> ### Author Response · Authors · 2022-11-15
> **Official Response to Reviewer yLyi**
>
> We thank the reviewer for the positive comments and valuable feedback. We have incorporated the comments and suggestions in the updated version of the paper.
>
> **Q**:  My main question is how to use counterfactual invariance in the learning process. As far as I understand, the optimization object is Eq. 2, where the authors want to obtain a model to obtain $\hat Y$ which is conditional independent of  $A$ given $Z$. And the realization of HSCIC in Eq. 2 is Eq. 3. But Eq. 3 seems to be estimating the conditional independence between $Y$ and $A$ given $Z$, where $Y$ is the true label in the sample. Shouldn't they have been conditional independent because you assume there is a pre-known causal graph and in the causal graph they are conditional independent? Why isn't it $\hat H_{\hat Y,A|Z}$, where $\hat Y$ denotes the predicted label of the training model?
>
> **A**: You are absolutely right. In the formula in Eq. 3, we report on the general approximation formula. However, in the experiments, we would use $H_{\hat{Y},A|Z}$. We changed the paper accordingly to make the paper clearer on this point (see Eq. 3).
>
> **Q**:  Interventional distribution differs in general from the conditional distribution due to possible unobserved confounding effects. It is right. But "possible unobserved confounding effects" is just one possible reason. In fact, they can differ even if there are only observed confounding effects. Hence I suggest another statement.
>
> **A**: Thank you for the comment. We reformulated this statement in our paper at the beginning of Section 2.

---

### Official Review · Reviewer_zfVq · 2022-10-24

**Confidence:** 3
**Correctness:** 4
**Technical Novelty And Significance:** 3
**Empirical Novelty And Significance:** 1
**Recommendation:** 5

**Clarity, Quality, Novelty And Reproducibility:**

The work is largely based on that of [Veich et. al (2021)], but the technical contributions of the new definition/theory should be acknowledged. The presentation is clear. The overall quality is good, but with some key missing information as I mentioned in the previous preview. I did not look into the code and thus cannot comment on the reproducibility.

**Strength And Weaknesses:**

Strength:
1. The notion of counterfactual invariance and the corresponding theory of the proposed method is a nice improvement over that of [Veitch et al. (2021)].
2. The proposed method is very intuitive.
3. The presentation of this paper is clear, organized, and easy to follow.
4. The literature review is thorough, with connections and improvements clearly stated.

Weakness:
1. I wonder how practical it is in reality to use the proposed method. For example, the estimation of HSCIC seems computationally expensive, what would be the extra computational cost for the added the regularization term?
2. I understand that the proposed method has theory guarantees for counterfactual invariance which many heuristic method does not possess. However, in practice, does it really outperform those heuristic methods? I personally think this is very important and should be clearly answered in the paper (the answer does not necessarily have to be a "yes", but a clear comparison should at least be presented). For example, in the experiments, I would expect to see some comparison with those well-known methods like data augmentation. However, the current experiment section seems limited, and does not contain any such information. I would encourage authors to add more informative experiments to convince readers like myself that we should actually use the proposed method instead of the other famous heuristics we've been used to.
3. Also, as mentioned by the authors themselves, the assumption of a known causal graph seems very strong, and could be another potential reason to prevent people from actually using the proposed method.

**Summary Of The Paper:**

This paper proposes a method to learn predictors that are invariant under counterfactual changes of certain covariates. The proposed method adds a model-agnostic regularization term based on the notion of conditional independence and kernel mean embeddings, to enforce the so called "counterfactual invariance" during training phase. Theoretical soundness of the proposed method is proved. Empirical experiments on both synthetic and real-word data sets are presented as well.

**Summary Of The Review:**

The proposed method and corresponding theory of this work is a nice and significant improvement over the work of [Veich et al. (2021)]. However, it misses some key experiments which greatly affects the practicality of the proposed method.

---

> ### Author Response · Authors · 2022-11-15
> **Official Response to Reviewer zfVq**
>
> We thank the reviewer for their time and insightful feedback.  We have incorporated the comments and suggested experiments in the updated version of the paper.
>
> **Q**: I wonder how practical it is in reality to use the proposed method. For example, the estimation of HSCIC seems computationally expensive, what would be the extra computational cost for the added regularization term?
>
> **A**: In a dataset with $n=1000$ data points, in the same structure of the first experimental setting shown in the paper (Figure 3), the average running time for an epoch without the regularization term is 0.003s and 1.112s with the trade-off term. In these results, mini-batch training with SGD with batch-size of 512 was used. By using smaller batch sizes, e.g. n=128, the extra computational cost can further decrease. From a theoretical perspective, the estimation of the HSCIC requires kernel ridge regression (see eq. (1) in our submission). In the *high-dimensional image* example, with mini batch-size of 512, the average running time for an epoch with the trade-off term is 64.03s and 34.01s without. This operation generally requires $O(n^3)$ and $O(n^2)$ memory, with $n$ the size of the dataset. However, this bound can be significantly improved by using, i.e., Fourier Features (see, i.e., [1,2]). By using Fourier features, the resulting approximate kernel ridge regression estimator can be computed in $O(ns^2)$ and $O(ns)$ memory. Here, $s$ is a parameter determining the accuracy of the approximation. In practice, $s$ can be set to be significantly smaller than the problem size, resulting in a dramatic **speed-up**. Other methods for *efficient* kernel computation include the popular Nyström approximation [3,4], and Memory-Efficient Kernel Approximation (MEKA) [5].
>
> **Q**: I understand that the proposed method has theory guarantees for counterfactual invariance which many heuristic methods do not possess. However, in practice, does it really outperform those heuristic methods?  I personally think this is very important and should be clearly answered in the paper (the answer does not necessarily have to be a "yes", but a clear comparison should at least be presented).
>
> **A**: Heuristic methods like data augmentation often fail to generate data consistently with the true data-generating process, especially when the true graph is complex or unknown. However, in order to evaluate counterfactuals, this is actually a necessary step. Differently, if heuristic methods are based on domain knowledge, for instance considering a known DAG, then our method has theoretical guarantees for counterfactuals, different from many heuristic methods. We appreciate the suggestions and we added an empirical comparison of data augmentation-based heuristic methods for counterfactual invariance in **Appendix E.7**.  The results of this empirical comparison are shown in the following table:
>
> Table 6: Results of **MSE** and **VCF** (all times $10^2$ for readability) on synthetic data of our method with trade-off parameters $\gamma=\tfrac{1}{2}$ and $\gamma=1$ with the heuristic methods *data augmentation* and *causal-based data augmentation*.
> ||**VCF**|**MSE**|
> |-----------| ----------- | ------------ |
> |data augmentation| $3.12 \pm 0.16$ |$0.003 \pm 0.001 $
> |causal-based data augmentation|$3.04 \pm 0.16$ | $0.013 \pm 0.012$ |
>  |$\gamma=\tfrac{1}{2}$  | $2.80 \pm 0.13$ |$0.044 \pm 0.022 $  |
>  | $\gamma=1$   | $2.34 \pm 0.19$ | $0.19 \pm 0.072  $  |
>
> **Q**: The assumption of a known causal graph seems very strong, and could be another potential reason to prevent people from actually using the proposed method.
>
> **A**:  We agree that knowing the causal graph is a strong assumption. However, essentially *all* work on cause-effect estimation and causal invariance starts from this assumption. In fact, while the somewhat orthogonal field of causal discovery aims at inferring the causal graph from (observational and/or interventional) data, most other work on causality starts from assuming some background knowledge of the data-generating process, we are aware that this is indeed a potentially problematic assumption. Usually, one refers to “expert knowledge” for how to come up with the graph. Ultimately, dismissing our work on the grounds that the true causal graph is hard to know in practice, is essentially dismissing large parts of the work on causality in machine learning altogether.
>
> [1] Ali Rahimi, Benjamin Recht: Random Features for Large-Scale Kernel Machines. NIPS 2007
>
> [2] Haim Avron, et al.: Random Fourier Features for Kernel Ridge Regression: Approximation Bounds and Statistical Guarantees. ICML 2017: 253-262
>
> [3] Petros Drineas, Michael W. Mahoney: On the Nyström Method for Approximating a Gram Matrix for Improved Kernel-Based Learning. JMLR 6: 2153-2175 (2005)
>
> [4] Cho-Jui Hsieh, et al.: Fast Prediction for Large-Scale Kernel Machines. NIPS 2014: 3689-3697
>
> [5] Si Si, et al.: Memory Efficient Kernel Approximation. JMLR 18: 20:1-20:32 (2017)

---

### Official Review · Reviewer_RdaH · 2022-10-26

**Confidence:** 2
**Correctness:** 3
**Technical Novelty And Significance:** 2
**Empirical Novelty And Significance:** 2
**Recommendation:** 6

**Clarity, Quality, Novelty And Reproducibility:**

The paper is well written, and the organization is comfortable. The paper is not quite novel, and the novelty comes from applying Kernel mean embeddings to an independent test.

**Strength And Weaknesses:**

In general, the paper is well written. The studied problem is interesting, which can be used in a lot of scenarios, for example fairness and so on. The proposed model is interesting, and the theory seems to be solid. My concerns come from two point: the first one is that, why we must use Kernel mean embeddings as the tool to test independence. Whether can we use other techniques? If there are advantages, please detail them to make this paper model compact. The second concern is about the experiment, fairness should be a very common direction. There should be more baselines, without enough baselines, it is hard to say the proposed model is effective.

**Summary Of The Paper:**

This paper aims to learn predictors that are invariant under counterfactual changes of certain covariates. To achieve this goal, the authors design a counterfactual invariant predictor. The implementation is highly based on the Kernel mean embeddings and conditional measures. By deriving the sufficient condition of the independence between Y and A, the authors propose an optimizable target to learn the independence. In the experiments, many experiments are conducted to verify the model effectiveness.


**Summary Of The Review:**

See the above comments.

---

> ### Author Response · Authors · 2022-11-15
> **Official Response to Reviewer RdaH**
>
> We thank the reviewer for the useful comments, pointing out that the paper is well-written and the problem is interesting. We appreciate the questions on kernel mean embeddings and fairness experiments and we are happy to reply to them below.
>
> **Q**: Why must we use Kernel mean embeddings as the tool to test independence? Can we use other techniques instead?
>
> **A**: First of all, we would like to clarify that our work does not involve independence testing, but rather the use of HSCIC as a conditional dependency measure. For this purpose, other techniques can also be used. We adopt the kernel mean embedding (KME) framework because it allows us to capture nonlinear statistical dependencies in the data without parametric assumptions about the underlying probability distribution, which gives our approach more flexibility. It is also known to work well for high-dimensional data. Furthermore, thanks to kernel methods, our approach can deal with not only data in a Euclidean space, but also continuous data and other structured data such as sequences, graphs, and distributions.
>
> **Q**: The second concern is about the experiment. Fairness is a very common direction. There should be more baselines. Without enough baselines, it is hard to say the proposed model is effective.
>
> **A**: As you correctly observed, algorithmic fairness is a very common and relevant direction for counterfactual invariance. However, in our work, we emphasize that this is not the only one, as there are other applications such as image classification on which we focus on. In our experimental settings, we compare our methods with fairness baseline methods and analyze the performance on fairness-related real-world baseline datasets. In our experiments (subsection: Comparison with baselines, Table 1) we empirically compare our method to the Counterfactual Fairness method by Kusner et al., the work that introduced the notion of counterfactual fairness. In terms of baseline datasets, we applied our method to the UCI Adult Dataset in Section 4.3. Even though a major hurdle in further research into fair learning is the lack of gold standard benchmarks, the Adult dataset is often used for empirical evaluation.

---

### Official Review · Reviewer_Udyy · 2022-11-01

**Confidence:** 3
**Correctness:** 3
**Technical Novelty And Significance:** 3
**Empirical Novelty And Significance:** 2
**Recommendation:** 5

**Clarity, Quality, Novelty And Reproducibility:**

Paper is clearly written but novelty is hard to assess without additional clarifications as several aspects of the proposed method exist in prior work.

**Strength And Weaknesses:**

Paper is well written and the proposed method is a sound way of obtaining counterfactual-invariant predictors given a set of observed adjustment variables under a known causal graph. It is also interesting that one does not need identifiability of the exact counterfactual distributions from observational data in order to learn general counterfactual-invariant predictors (as in Theorem 3.2).

I have the following major concerns:

1. The comparison with existing notions of counterfactual invariance is incorrect, possibly due to the use of different notations. In [1], Veitch et al. define Y(a) using a counterfactual change A=a “leaving all else fixed” including the exogenous variables, thus giving the true counterfactual for that observation. The current paper claims that the counterfactual-invariance definition in [1] only implies invariance to interventions Y|do(a) , which is incorrect. While the practical method proposed in [1] cannot differentiate between interventions and counterfactuals (as discussed in Section 3 of [1]), the definition itself is general.

   Further, a definition of counterfactual-invariance with a clearer notation is provided in [2] that explicitly shows conditioning of the observed variable. This definition is more general than the one  proposed in the current paper as it requires almost-sure equality of representations (and not just in-distribution equality).

2. Experimental evaluation:

   1. All the experiments consider the case when A (protected attribute) directly causes the target Y. This seems to be in contradiction with Theorem 3.2 which requires that a set of nodes Z blocks all paths from A to Y. In experiments, the original counterfactual fairness [3] setting should also be considered where A does not directly cause Y.
   2. Additional baselines: Since one of the contributions of the paper is to use HSCIC for conditional independence testing, other more classical approaches (e.g., [4,5]) for the same should be compare against. Further, since the current experiments with A directly causing Y are more suited to path-specific counterfactual fairness [6], this should be added as baseline whenever possible.
   3. Authors should provide a method/heuristic to select the regularization parameter $\gamma$ using only the training/validation data.

3. I am not able to appropriately judge the novelty of the proposed method.

   1. Authors should discuss more in detail how Theorem 3.2 and its proof generalizes the work in [7,8]. For example, the construction of counterfactual graph in the proof of Theorem 3.2 is claimed to be more general than the one in [8], but not elaborated further.
   2. A different definition of HSCIC (functionally equivalent to [9] with same implementation) is introduced with one fewer restriction but it is unclear from the text how the reduced assumption impacts the current task. For instance, do the current tasks exhibit violations to the original stronger assumption in [9]?


Other questions/comments:

1. Please discuss/refer Figure 1(b-c) in the main text.
2. Please use the same precision for numeric values in Table 1. Also, indicate what value of hyperparameter $\gamma$ is chosen based just on the training data (without looking at test performance) for a fair comparison with the baselines.
3. In Table 2, with dimA=2 and $\gamma=1$, VCF strangely increases but HSCIC decreases. Is there a reason for this anomalous behavior?


**References**

[1] Victor Veitch, Alexander D’Amour, Steve Yadlowsky, and Jacob Eisenstein. Counterfactual invariance to spurious correlations in text classification. In Advances in Neural Information Processing Systems, 2021

[2] Mouli, S. Chandra, and Bruno Ribeiro. "Asymmetry learning for counterfactually-invariant classification in ood tasks." *International Conference on Learning Representations*. 2022.

[3] Kusner, Matt J., et al. "Counterfactual fairness." *Advances in neural information processing systems* 30 (2017).

[4] Zhang, Kun, et al. "Kernel-based conditional independence test and application in causal discovery." *arXiv preprint arXiv:1202.3775* (2012).

[5] Doran, Gary, et al. "A Permutation-Based Kernel Conditional Independence Test." *UAI*. 2014.

[6] Chiappa, Silvia. "Path-specific counterfactual fairness." *Proceedings of the AAAI Conference on Artificial Intelligence*. Vol. 33. No. 01. 2019.

[7] Ilya Shpitser and Judea Pearl. Complete identification methods for the causal hierarchy. Journal of Machine Learning Research, 9:1941–1979, 2008.

[8] Ilya Shpitser and Judea Pearl. Effects of treatment on the treated: Identification and generalization. In Proceedings of the Twenty-Fifth Conference on Uncertainty in Artificial Intelligence,, pp. 514–521, 2009.

[9] Junhyung Park and Krikamol Muandet. A measure-theoretic approach to kernel conditional mean embeddings. In Advances in Neural Information Processing Systems, pp. 21247–21259, 2020.





**Summary Of The Paper:**

The paper proposes a method to obtain predictors that are counterfactually-invariant to certain observed variables. Authors first show that counterfactual invariance of a predictor with respect to the variables A can be ensured given a set of variables Z that block all paths from A to Y. That is, a counterfactual invariant predictor satisfies the conditional independence $\hat{Y} \perp A | Z$. Then, the authors propose a conditional independence criterion and show that the proposed approach can learn counterfactual invariance in different tasks.

**Summary Of The Review:**

Comparison with existing notions of counterfactual invariance is incorrect and the claims of generality with respect to these existing notions is not well-supported. Problems in empirical evaluation: experiments do not seem to follow the assumptions in the main theorem, missing relevant baselines and missing procedure to choose $\gamma$, an important regularization hyperparameter.

---

> ### Author Response · Authors · 2022-11-15
> **Official Response to Reviewer Udyy (1/2)**
>
> We thank the reviewer for your insightful feedback, pointing out that the method is well-written and that the method is sound. We also see that you have concerns and questions. We have incorporated the comments and suggestions in the updated version of the paper.
>
> **Q**: The comparison with existing notions of counterfactual invariance is incorrect, possibly due to the use of different notations.
>
> **A**: It seems to us that there has been a misunderstanding. In our paper, we do not claim that our notion of counterfactual invariance is equivalent to the one proposed by [1]. In particular, we do not claim that the counterfactual-invariance definition in [1] only implies invariance to interventions Y|do(a). In fact, we are well aware that the setting studied by [1] is stronger than ours. In Section 3, we simply observe that our method can be applied to the causal and anti-causal structure studied by [1], with the advantage that our proposed method entails **sufficient** conditions. Furthermore, we observe in specific causal structures the two methods converge in practice, while in other experimental settings their performance is comparable (see Appendix E.6). After re-reading our submission, we realized that some sentences in the paragraph before Corollary 3.7 can be misleading. We improved the write-up to avoid misunderstandings and clarified this point after corollary 3.7.
>
> **Q**: Further, a definition of counterfactual-invariance with a clearer notation is provided in [2] that explicitly shows conditioning of the observed variable.
>
> **A**: Yes, thank you for pointing to this bibliographical reference, showing that counterfactual invariance can be used for learning an invariant classifier that generalizes well in held-out examples from the training and test distributions. We have cited it in our updated manuscript. We remark, however, that in our paper we provide a general graphical criterion for invariance, based on observed variables only. To the best of our knowledge, we are the first ones to provide a criterion as in Theorem 3.2, and use KMEs to enforce this criterion. Our experiments confirm that our method is suitable for a variety of applications, including the important application of counterfactual fairness.
>
> **Q**: All the experiments consider the case when $A$ (protected attribute) directly causes the target $Y$. This seems to be in contradiction with Theorem 3.2 [...].
>
> **A**: We think this doubt is due to a terminology misunderstanding. Theorem 3.2 states that the set of nodes $Z$ blocks all **non-directed** paths from $A\cup W$ to $Y$. The notion of non-directed paths in the DAG literature is equivalent to the one of non-causal paths in the causality one.  Based on the definition of non-causal path given in *On the Validity of Covariate Adjustment for Estimating Causal Effects (I. Shpitser et al., 2012): ‘A path from a node $X$ to a node $Y$ consists exclusively of directed arrows pointing away from $X$ is called directed or causal, all other kinds of paths are called non-causal.’* Based on this definition, the direct edge $A\to Y$ is not a non-directed causal path between $A$ and $Y$ and hence does not need to be blocked by $Z$ for Theorem 3.2 to hold. We understand the confusion and we apologize if the text was not clear. In order to make the notation clearer and to avoid possible misunderstandings, we have modified the terminology of non-directed path as non-causal path in the text and we added a definition of non-causal paths in Theorem 3.2.
>
> **Q**: Additional baselines: Since one of the contributions of the paper is to use HSCIC for conditional independence testing, other more classical approaches (e.g., [4,5]) for the same should be compared against.
>
> **A**: Unlike [4,5], we do not use HSCIC for conditional independence *testing*, but rather as a conditional dependency measure. We apologize for this confusion. We have now provided a clarification in the updated manuscript that our work does not use HSCIC for conditional independence testing (Appendix C.1).
>
> **Q**: Authors should provide a method/heuristic to select the regularization parameter using only the training/validation data.
>
> **A**: We emphasize that $\gamma$ is not a regularization parameter. Instead, it should be viewed as a tunable knob that allows one to trade-off between predictive performance and counterfactual invariance. The optimal value of $\gamma$ depends on the desired predictor properties in a given situation/application scenario. Hence, it cannot be identified from the training data alone.

---

> > ### Author Response · Authors · 2022-11-15
> > **Official Response to Reviewer Udyy (2/2)**
> >
> >
> > **Q**: I am not able to appropriately judge the novelty of the proposed method. Authors should discuss more in detail how Theorem 3.2 and its proof generalizes the work in [7,8].
> >
> > **A**: Our proof is based on the observation that the set $Z$ as in Theorem 3.2 acts as a $d$-separator for certain random variables in a graph that allows reasoning about dependencies among pre- and post-interventional random variables. This graph simplifies the counterfactual graph by [7], and it generalizes the augmented graph structure described in Theorem 1 by [8]. We can then combine this property with covariate adjustment to prove the claim. Our proof does not rely on the identification of the counterfactual distributions (e.g., by simply applying the do-calculus). In particular, the assumptions do not rule out hidden confounding in the model. We added a clarification in the updated manuscript.
> >
> > **Q**: A different definition of HSCIC (functionally equivalent to [9] with the same implementation) is introduced with one fewer restriction but it is unclear from the text how the reduced assumption impacts the current task.
> >
> > **A**: The advantage is mostly theoretical and it does not impact the current task in practice. In [9] the authors prove that $\text{HSCIC}(X, Y | Z) = 0$ almost surely if and only if $X \perp Y | Z$, under the assumption that the underlying measurable spaces admit a regular version (see Theorem 5.4 by [9]). The existence of a regular version is essential in the proof given by [9], since their definition of the HSCIC is based on the Bochner conditional expected value. This assumption, however, is a rather technical measure-theoretic condition (see Definition 2.7 and 2.8 by [9]). It is unclear which kernels fulfill this condition. By contrast, our definition of the HSCIC does not use the Bochner conditional expected value, but it is defined by KMEs of conditional measures directly. Due to this different approach, our conditional dependency measure does not require the existence of a regular version, and it applies to any characteristic kernel.
> >
> > **Q**: Please discuss/refer Figure 1(b-c) in the main text.
> >
> > **A**: Yes, thank you for your comment. The causal diagrams in Figure 1(b)-1(c) are mainly interesting since they are studied by [1]. We discuss them in Appendix E.6. We added a reference to these diagrams in Section 3, before Corollary 3.7.
> >
> > **Q**: Please use the same precision for numeric values in Table 1. Also, indicate what value of hyperparameter is chosen based just on the training data. The goal of this comparison is to show that our method can be tuned to reach a higher level of counterfactual invariance compared to the shown baselines, while achieving a competitive prediction accuracy.
> >
> > **A**: Yes, we gladly fixed this inaccuracy. As previously mentioned, we do not choose the values of the hyperparameter based on training data here.
> >
> > **Q**: In Table 2, with dimA=2 and, VCF strangely increases but HSCIC decreases. Is there a reason for this anomalous behavior?
> >
> > **A**: Thank you a lot for noticing this typo. This number is actually 0.022.

---

### Comment · Program_Chairs · 2022-11-21
**Sharing an anonymous review and counter example for reviewers and AC to discuss**

An anonymous reviewer contacted the PC and claimed that he/she has reviewed this submission in a prior conference and identified a major technical flow that remains unaddressed. The PCs are requested to paste the comments below:

Consider the following causal model:
T->X->Y
Now using the notation of theorem 3.2 let A={T}, W={T,X} and Z be the empty set. As there are no non causal paths between A \union W to Y, Z trivially blocks all paths. Therefore theorem 3.2 says if we have \hat{Y} \perp A that \hat{Y} is counterfactually invariant in A with respect to W, so that P(Y(t)=y | T=t,X=x)=  P(Y(t')=y | T=t,X=x). This would mean every predictor that satisfies demographic parity with respect to this model satisfies counterfactual fairness, however this is not true. Consider the following:

N_1,N_2 are independent Bernoulli with probability 1/2. Let T=N_1, X=(T)*(N_2)+ (1-T)*(1-N_2) and Y,\hat{Y}=X. Now under this model X is independent of T and therefore so is \hat{Y}. But we have that Y(t) given X=x, T=t is x but Y(t') for t' not equal to t given X=x, T=t is 1-X and so the two are in fact never equal. Therefore this model is a simple counterexample to theorem 3.2. Gaussian noise could be added to make this example non deterministic.

---

> ### Author Response · Authors · 2022-11-22
> **Reply to anonymous review**
>
> We would like to thank the PC for sharing the anonymous review.
>
> Our theorem does not imply that counterfactual fairness equals demographic parity. In fact, for any SCM where there are non-directed paths between the protected attribute and the outcome, our graphical criterion does not imply that the two conditions are equivalent. Also, the counterexample is incorrect. Since Y, \hat{Y} = X then Y, \hat{Y} this causal diagram reduces to T->X, and counterfactual invariance then becomes P(X(t)=x | T=t,X=x)= P(X(t’)=x | T=t,X=x). However, our theorem does not apply to this case, since our graphical criterion applies if the outcome is not part of the conditioning set.
>
> We are happy to improve our paper by adding the explicit condition that our set W must be disjoint from the outcome variable. We are currently working on an improved, clearer version of our theorem.

---

> > ### Comment · Area_Chair_KdPf · 2022-12-01
> > **Please share theorem changes when ready**
> >
> > Dear Authors,
> >
> > Please share here all changes when ready. I will help our discussion.
> >
> > Thanks,.
> > AC

---

> > > ### Author Response · Authors · 2022-12-07
> > > **Changes to the theorem**
> > >
> > > To address the concern of the anonymous reviewer, we can modify the theorem as follows:
> > >
> > > Theorem 3.2: Let G be a causal diagram, and let A, W be two sets of nodes in G, such that (A ∪ W) ∩ Y = ∅. Let Z be a set of nodes that blocks all non-causal paths from A ∪ W to Y. Then, for any SCM compatible with G, any predictor hat{Y} conditionally independent of A given Z, is counterfactually invariant in A with respect to W.
> > >
> > > This modification addresses the incorrect counterexample. However, we found that there is a bug in the current version of this manuscript, since the set Z in Theorem 3.2 should consist of a larger set of random variables in general, including a set of variables that fulfill a valid adjustment criterion similar to the current version of the theorem. Although Theorem 3.2 in its current form is not technically incorrect in some cases, we will have to re-run some of the experiments to fix this problem. Re-running all experiments will require a few weeks of work. However, we have run again the experiments on the synthetic dataset (see Section 4 Figure 3 in our manuscript) and we observe no significant difference in performance. The results for this set of experiments are provided in the following table:
> > >
> > > | Gamma   |    MSE (x 10E+5)   |    HSCIC (x 10E+2)   |    VCF (x 10E+2)  |
> > > |---------|----------------|---------|----------------|
> > > | 0.0       |      1.0  ± 0.02     |       2.953 ± 0.028    |        3.102 ± 0.010 |
> > > | 0.1      |      6.3  ± 0.28       |     2.843 ± 0.025     |       2.981 ± 0.011 |
> > > | 0.2        |    26   ± 1.35       |      2.709 ± 0.024     |       2.864 ± 0.014|
> > > | 0.3         |   63   ± 2.78      |      2.541 ± 0.023      |      2.791 ± 0.012|
> > > | 0.4     |       110 ± 3.61      |      2.406 ± 0.024      |      2.662 ± 0.015|
> > > | 0.5      |      174 ± 3.34      |      2.240 ± 0.022      |      2.651 ± 0.017|
> > > | 0.6       |     234 ± 3.26      |      2.123 ± 0.021      |      2.550 ± 0.015|
> > > | 0.7      |      316 ± 5.34       |     2.001 ± 0.024      |       2.467 ± 0.019|
> > > | 0.8      |      407 ± 6.60     |       1.866 ± 0.021      |      2.347 ± 0.026|
> > > | 0.9     |       485 ± 7.00     |       1.755 ± 0.017      |      2.291 ± 0.020|
> > > | 1.0     |       566 ± 9.83     |       1.755 ± 0.014      |      2.122 ± 0.027|

---

### Decision · Program_Chairs · 2023-01-20

**Decision:**

Reject

**Justification For Why Not Higher Score:**

The work has some bugs that could not be fixed during rebuttal. Authors seem to be currently trying to address them.

**Justification For Why Not Lower Score:**

N/A

**Metareview: Summary, Strengths And Weaknesses:**

TL;DR: The work could not be fixed during rebuttal. There are many issues with the original submission (missing important prior works, limited comparisons, not a good justification for the chosen independence criteria, errors in the theory). The authors did their best to address them but ultimately the draft needs to have more significant modifications not suited for a rebuttal (it is better to resubmit).

This work proposes a method to obtain predictors that are counterfactually-invariant to certain observed variables. Authors first show that counterfactual invariance of a predictor with respect to the variables A can be ensured given a set of variables Z that block all paths from A to Y. That is, a counterfactual invariant predictor satisfies the conditional independence. Then, the authors propose a conditional independence criterion and show that the proposed approach can learn counterfactual invariance in different tasks.

Reviewers agree that the work is promising and the tasks are of value to the ML community. However, the work is not yet ready for publication. I strongly encourage the authors to take the good feedback they received into consideration and update their draft for a resubmission.

**Summary Of Ac-Reviewer Meeting:**

- Not all reviewers participated in the virtual meeting. We had reviewers in very different time zones and not all reviewers replied to the Doodle poll. A reviewer that was present at the meeting (a reviewer with a 5 score) said the paper had not originally done a good job at citing existing important work. The authors response was not convincing because it is still not clear why HSCIC is the independence criteria to be used and rather than some other independence criteria. There is not experimental comparison with alternative methods.